# 🤖 *Presenting a Paper is an Art*: SELF-IMPROVEMENT AESTHETIC AGENTS FOR ACADEMIC PRESENTATIONS

**Chengzhi Liu**[1]*, **Yuzhe Yang**[1]*, **Kaiwen Zhou**[2], **Zhen Zhang**[1], **Yue Fan**[2], **Yanan Xie**[3],
**Peng Qi**[3], **Xin Eric Wang**[1]
[1]University of California, Santa Barbara    [2]University of California, Santa Cruz    [3]Uniphore
{chengzhi,yuzheyang,ericxwang}@ucsb.edu

**Project Page:** https://evopresent.github.io/

## ABSTRACT

The promotion of academic papers has become an important means of enhancing research visibility. However, existing automated methods struggle limited storytelling, insufficient aesthetic quality, and constrained self-adjustment, making it difficult to achieve efficient and engaging dissemination. At the heart of those challenges is a simple principle: *there is no way to improve it when you cannot evaluate it right*. To address this, we introduce **EvoPresent**, a self-improvement agent framework that unifies coherent narratives, aesthetic-aware designs, and realistic presentation delivery via virtual characters. Central to EvoPresent is **PresAesth**, a multi-task reinforcement learning (RL) aesthetic model that provides reliable aesthetic scoring, defect adjustment, and comparative feedback, enabling iterative self-improvement even under limited aesthetic training data. To systematically evaluate the methods, we introduce **EvoPresent Benchmark**, a comprehensive benchmark comprising: *Presentation Generation Quality*, built on 650 top-tier AI conference papers with multimodal resources (slides, videos and scripts) to assess both content and design; and *Aesthetic Awareness*, consisting of 2,000 slide pairs with varying aesthetic levels, supporting joint training and evaluation on scoring, defect adjustment, and comparison. Our findings highlight that (i) High-quality feedback is essential for agent self-improvement, while initial capability alone does not guarantee effective self-correction. (ii) Automated generation pipelines exhibit a trade-off between visual design and content construction. (iii) Multi-task RL training shows stronger generalization in aesthetic awareness tasks.

## 1 INTRODUCTION

As scholarly communication increasingly moves online, the promotion of academic papers has become a crucial means of enhancing research visibility. Among various formats, presentation videos stand out for their efficiency and intuitiveness. Despite recent progress in automatic generation, including paper-to-slide (Ge et al., 2025; Zheng et al., 2025) generation and text-to-video synthesis (Shi et al., 2025; Xue et al., 2025), existing methods still present several notable limitations. Specifically, current approaches, as illustrated in Figure 1 (b–c), suffer from limited narrative coherence due to direct extraction; restricted design flexibility from fixed templates; and the absence of self-improvement mechanisms that leave systems overly dependent on manual intervention in academic communication. Second, presentation aesthetic evaluation remains underdeveloped, with existing methods lacking both adequate aesthetic awareness (Zhou et al., 2024b) and dedicated metrics, which limits the comprehensiveness and reliability of evaluation. Moreover, current evaluation methods largely rely on VLM-as-judge methods (Hwang et al., 2025), further reducing their consistency and robustness.

To address the limitations of generation quality, we propose **EvoPresent**, as shown in Figure 1(a), a self-improving agent framework that follows a draft–feedback–refinement iterative loop to produce academic presentations with both narrative coherence and aesthetic awareness. First, the *Storyline*

---

*Equal contribution

Figure 1: **Comparison between EvoPresent and other methods.** (a) EvoPresent achieves high quality with fewer iteration through its self-improvement framework, supporting multiple formats (videos, scripts, slides) for a more realistic presentation. (b) PPTAgent (Zheng et al., 2025) and PresentAgent (Shi et al., 2025) lack content expressiveness and are limited by fixed templates. (c) Paper2Poster (Pang et al., 2025) lacks flexibility and an effective visual checker, leading to poor visual design and requiring extensive adjustments.

*Agent* extracts core text and figures from papers to construct structured scripts. Next, the *Scholar Agent* expands the narrative and enriches content through tool selection (e.g., image generation), while the *Design Agent* handles layout design and visual rendering to produce slides and video frames. Finally, the *Checker Agent* evaluate the presentation's content and design while providing targeted feedback until it reaches the desired visual standard. In this process, the framework integrates **PresAesth**, a multi-task reinforcement learning-based aesthetic model to support the agents' iterative self-improvement. Trained on limited aesthetic preference data, PresAesth leverages Group Relative Policy Optimization (GRPO) (Shao et al., 2024) for joint multi-task learning. It consistently performs aesthetic scoring, defect correction, and pairwise comparison, ensuring reliable aesthetic perception and reasoning for the system. Through this framework, EvoPresent not only enhances narrative coherence and expressiveness but also achieves aesthetic-aware design, ultimately producing higher-quality and more engaging academic presentation.

To address the shortcomings of existing evaluation, we propose **EvoPresent Benchmark** (Sec. 4), which integrates two components: (i) *Presentation Generation Quality*, covering 650 academic resources across multiple domains and formats (slides, videos and scripts) with human annotations. This combines both global and fine-grained evaluations: the former focuses on overall narrative coherence and aesthetics, quantified with objective metrics (e.g., perplexity), while the latter leverages VLM-as-Judge to assess content and design across eight dimensions. (ii) *Aesthetic Awareness*, a dataset with 2000 slide pairs constructed with controlled perturbations (e.g., layout change) to provide a systematic framework for training and evaluation in aesthetic perception and reasoning.

We further conduct a systematic evaluation of EvoPresent against state-of-the-art models and multi-agent approaches, yielding the following key findings: (i) Multi-task RL demonstrates superior generalization in aesthetic awareness compared to other training paradigms. (ii) High-quality feedback proves essential for the iterative self-improvement of agent frameworks, yielding fewer iterations and faster progress. (iii) Agents' initial performance does not necessarily correlate with its correction ability, and high performance alone is insufficient to guarantee stronger correction. (iv) Automated generation tasks reveal an inherent trade-off between content construction and visual design, where aesthetic awareness emerges as the primary bottleneck. To sum up, our contributions are listed as follows:

- We introduce EvoPresent, the first self-improvement multi-agent framework for generating realistic academic presentations with minimal human intervention.
- We design PresAesth , a multi-task reinforcement learning aesthetic model that unifies scoring, defect adjustment, and comparison with limited human-preference aesthetic data.
- We propose EvoPresent Benchmark, a comprehensive framework that supports systematic evaluation of generation task performance and joint training–evaluation of aesthetic awareness tasks.
- Extensive experiments show that EvoPresent surpasses existing methods, delivering presentations of higher quality and engagement comparable to human-designed.

## 2 RELATED WORK

**Automated Presentation Generation.** Recent advances in multimodal large language models have driven progress in the automated generation of academic papers, including tasks such as academic

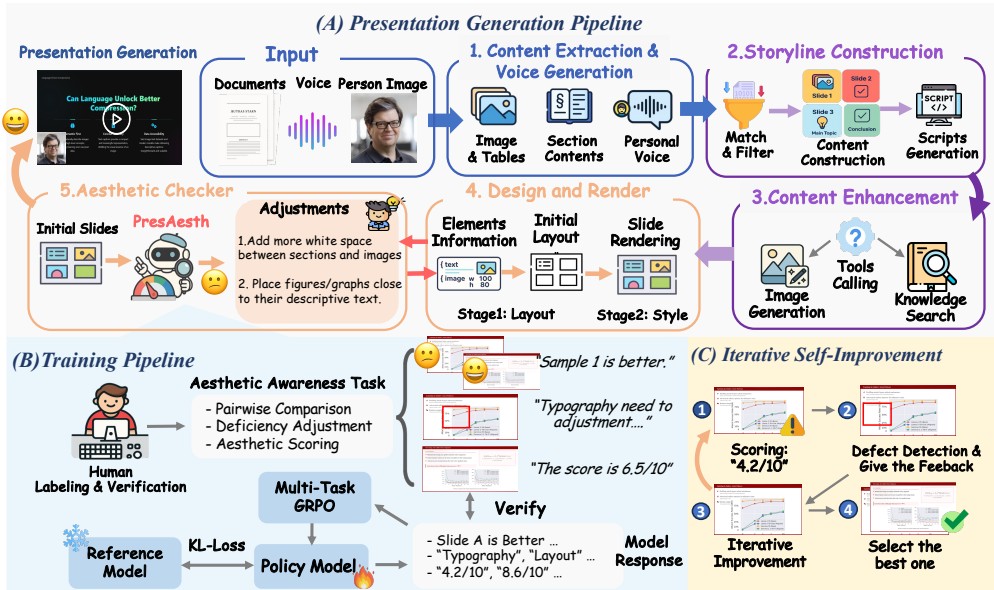

Figure 2: **Overview of the EvoPresent framework.** *(a)* EvoPresent first performs content extraction and voice generation, then constructs the storyline and script, followed by content enhancement using image generation and knowledge retrieval. Design and rendering are handled next, and the aesthetic checker evaluates the initial slide and provides adjustments. *(b)* PresAesth is trained on a human-preference aesthetic dataset via multiple tasks (scoring, defect adjustment, and comparison). *(c)* The PresAesth model guides the agent framework in iterative self-improvement.

slide creation (Zheng et al., 2025; Ge et al., 2025; Shi et al., 2025) and poster generation (Pang et al., 2025). However, the quality of the generated outputs often falls short of practical requirements. As shown in Figure 1(b), PPTAgent and PresentAgent primarily rely on direct content extraction and fixed templates, which results in limited narrative coherence. Figure 1(c) further indicates that while Paper2Poster employs a VLM-based checker, its limited aesthetic and design perception constrains generation quality. Moreover, most existing methods lack a reliable self-improvement mechanism, leaving the generation process heavily dependent on manual intervention (Li et al., 2023). In contrast, EvoPresent integrates storyline construction with iterative self-refinement, ensuring high-quality academic presentations with minimal refinement.

**Aesthetic Evaluation.** The aesthetic evaluation of visual design remains a formidable challenge, as existing VLMs still show a substantial gap in capturing the nuanced and subjective nature of aesthetics compared to human perception (Zhou et al., 2024b). Many methods are being explored, such as developing systems for accurately scoring image designs (Hong et al., 2023; Yu et al., 2021) and techniques for better aligning model outputs with human preferences (Zhou et al., 2024a; Li et al., 2025; Liao et al., 2025). However, existing methods are largely confined to natural image aesthetics and struggle with the higher subjectivity and complexity of academic scenarios (e.g., slides), leading to unstable performance. In contrast, our proposed PresAesth, trained via multi-task RL, exhibits stronger aesthetic awareness, enabling more reliable evaluation of academic visual design.

## 3 EVOPRESENT FRAMEWORK

**Overview.** We propose **EvoPresent**, a self-improvement agent framework built upon a draft–feedback–revision cycle. The pipeline leverages four agents in sequence: the *Storyline Agent* extracts key information from the paper to construct the storyline and script, the *Scholar Agent* enriches them with external knowledge and image generation tools, the *Design Agent* generates layouts and renders initial drafts of slides and video frames, and the *Checker Agent* evaluates content and design to provide targeted feedback for refinement. Especially, to overcome the challenge of aesthetic evaluation and ensure reliable adjustment, we integrate **PresAesth** (Sec. 3.2), a multi-task RL aesthetic model that performs three core tasks-scoring, defect correction and pairwise comparison, thereby supporting the agent framework's effective iterative self-improvement.

## 3.1 EVOPRESENT WORKFLOW

**Storyline Agent.** Given an paper, the first step involves organizing the information and conceptualizing the storyline. The Storyline agent first employs Marker (Paruchuri, 2025) to extract text and visual elements, storing them in a unified library. To minimize redundancy and ensure completeness, the agent performs several key functions: (i) constructs a story framework by dividing the paper into thematic sections; (ii) reorganizes the text into coherent chapters, extracting key arguments and details; (iii) associates pertinent visuals and formulas with the content while filtering out irrelevant elements; (iv) generates a complete presentation script based on the constructed storyline.

**Scholar Agent.** The Scholar Agent optimizes the paper's storyline to enhance both the academic content and visual quality of the presentation. It first analyzes the storyline to identify gaps in academic information and enriches it through: (i) Knowledge Enrichment: using tools such as the ArXiv Metadata and Citation Parser (MCP) (Blazick, 2025) to access additional related knowledge. (ii) Visual Enhancement: employing GPT-4o (OpenAI, 2024) and Qwen-Image (Wu et al., 2025) to generate relevant charts or diagrams, improving visual impact and aligning visuals with the text.

**Design Agent.** Once the storyline is defined, our Design Agent begins designing the presentation pages. This agent consists of two core components: the *Layout Planner* and the *Style Render*. The process begins with the *Layout Planner*, which addresses the crucial task of arranging elements with aesthetic precision. To overcome the instability of using an agent without visual feedback, the planner first generates an initial layout based on text and image sizes to ensure element balance and correct aspect ratios. With a stable layout in place, the *Style Render* renders the overall visual design. It analyzes the overall theme to select a suitable style from a predefined Cascading Style Sheets (CSS) library, applying only visual elements like colors, fonts, and backgrounds to ensure the layout remains flexible. Finally, the agent adjusts font sizes to create a clear information hierarchy and applies dynamic effects to enhance audience engagement. Both components directly impact the final quality, and our choice of HTML over the less stable PPTX format provides the necessary flexibility and aesthetic control for this workflow (see Appendix 12 for a format comparison).

**Checker Agent.** The Checker Agent is a critical component designed for iterative refinement of presentation. Integrated with the PresAesth model (Sec. 3.2), it performs stable aesthetic perception of slides. As shown in Algorithm 1, the agent evaluates the aesthetic score of the slide in each iteration, terminating early if the score exceeds the threshold (The default aesthetic threshold is set to 8.0; further details are provided in Appendix B.) and otherwise providing feedback to the Layout Planner for refinement. The agent further compares scores across iterations, reverting to the highest-performing version in case of quality degradation, and if no iteration meets the target score, the version with the highest score is selected as the final output. The finalized slides are then paired with narration audio generated by a text-to-speech system, with each audio segment temporally aligned to its relevant slide. The video presents slides for the duration of narration, optionally incorporating transitions. Through this iterative refinement process, the Checker Agent ensures both aesthetic consistency and stability. See Appendix F.1 for video generation details.

---

**Algorithm 1** Checker Agent Workflow

**Require:** Initial slides $S^{(0)}$, Layout Planner $L$, Checker Agent $C$, iterations $T$, threshold $S_{th}$
**Ensure:** Presentation video $V_{pre}$
1: $S_{best} \leftarrow S^{(0)}$, $\text{Score}_{best} \leftarrow 0$
2: **for** $t = 0$ **to** $T - 1$ **do**
3:      $\text{Score}^{(t)} \leftarrow C.\text{score}(S^{(t)})$    ▷ *Scoring*
4:      **if** $\text{Score}^{(t)} \geq S_{th}$ **then**
5:          **return** $V_{pre} \leftarrow \text{generate}(S^{(t)})$
6:      **end if**
7:      $U \leftarrow S^{(t)}$
8:      **if** $t > 0 \wedge \text{Score}^{(t)} < \text{Score}^{(t-1)}$ **then**
9:          $U \leftarrow S^{(t-1)}$    ▷ *Revert to better version*
10:     **end if**
11:     $\text{Feedback} \leftarrow C.\text{feedback}(U)$
12:     $S^{(t+1)} \leftarrow L.\text{refine}(U, \text{Feedback})$
13:     **if** $\text{Score}^{(t)} > \text{Score}_{best}$ **then**
14:        $S_{best} \leftarrow S^{(t)}$, $\text{Score}_{best} \leftarrow \text{Score}^{(t)}$
15:     **end if**
16: **end for**
17: **return** $V_{pre} \leftarrow \text{generate}(S_{best})$    ▷ *Select the best*

---

## 3.2 PRESAESTH: MULTI-TASK AESTHETIC AWARENESS MODEL

**Tasks Formulation.** Slides serve as the primary medium in academic presentations, and accurate perception and evaluation of their aesthetics is essential to overall quality. Accordingly, we define three core tasks for the PresAesth model: **(i)** Scoring: Given a single slide image as input, this task is to evaluate its absolute quality on a numerical scale, providing a holistic quantitative assessment. **(ii)** Defect Adjustment: Taking a single slide image as input, this task requires identifying specific deficiencies and providing specific feedback for improvement. Deficiencies are classified into three main categories: *Composition & Layout, Typography, and Imagery & Visualizations*. **(iii)** Comparison: Given a baseline slide image and two proposed revisions, the task is to identify which of the two

revisions offers a superior improvement over the baseline. Overall, these tasks jointly aim to endow the model with aesthetic awareness consistent with human preferences.

**Multi-Tasks GRPO.** To train PresAesth, we employ GRPO, an efficient RL method that leverages verified reward signals to capture the subjective nature of aesthetic evaluation, with Qwen-2.5-VL-7B (Bai et al., 2025) serving as the base model. Unlike Supervised Fine-Tuning (SFT), which fails to learn complex aesthetics from simplistic ground-truth labels. RL approach allows the model to explore and develop sophisticated reasoning by learning from verified reward signals. Consequently, our model delivers both accurate aesthetic judgments and coherent reasoning, enabling precise and effective feedback to steer our agentic workflow. GRPO's group-based optimization intrinsically matches how humans make aesthetic judgments through comparative assessment rather than absolute scoring. We apply the same loss function as in (Shao et al., 2024), details are in Appendix C.1.

To guide the training process, we design a comprehensive reward function. The first component is a Format Reward, which ensures the model's outputs are structured and parsable. We require the model to articulate its reasoning process within $\langle\texttt{think}\rangle$ and $\langle/\texttt{think}\rangle$ tags, and present its final conclusion within $\langle\texttt{answer}\rangle$ and $\langle/\texttt{answer}\rangle$ tags. A reward of 1 is issued if the response strictly adheres to this format, and 0 otherwise, encouraging the generation of well-organized outputs. The second component is an Accuracy Reward ($r_{\text{acc}}$), which evaluates the correctness of the content within the answer tag. This reward is task-dependent, formulated as:

$$r_{\text{acc}} = \begin{cases} \mathbb{I}(o_{\text{comp}} = y_{\text{comp}}) & \textit{for Comparison Task} \\ \mathbb{I}(\text{F1}(f(o_{\text{def}}), y_{\text{def}}) > \alpha) & \textit{for Adjustment Task} \\ \mathbb{I}(|o_{\text{score}} - y_{\text{score}}| < \zeta) & \textit{for Scoring Task} \end{cases}$$

where $\mathbb{I}(\cdot)$ is the indicator function; $o_{\text{comp}}$, $o_{\text{def}}$, and $o_{\text{score}}$ are the model's outputs for each task, while $y_{\text{comp}}$, $y_{\text{def}}$, and $y_{\text{score}}$ are the associated ground-truth labels. The function $f(\cdot)$ parses the model's textual feedback to extract deficiency categories. $\alpha$ and $\zeta$ are predefined tolerance thresholds for the F1-score and scoring error, respectively. The overall reward for the $i$-th response, $r^{(i)}$, is the sum of its format and accuracy rewards: $r^{(i)} = r_{\text{fmt}}^{(i)} + r_{\text{acc}}^{(i)}$. This reward structure provides distinct incentives for both correct formatting and factual accuracy. Training settings are detailed in Appendix C.2.

# 4 EVOPRESENT BENCHMARK

## 4.1 TASK DEFINITION

The EvoPresent benchmark consists of two core tasks: (i) Presentation Generation Quality, which evaluates both content and design dimensions; and (ii) Aesthetic Awareness, established as an integrated training–evaluation framework for aesthetic scoring, defect adjustment, and pairwise comparison. See Appendix B.1 for detailed benchmark settings.

**Generation Quality.** We evaluate the presentation generation quality using two dimensions: (i) *Global Evaluation*, which employs objective metrics to assess overall performance. For content, we measure coherence and fluency with Perplexity (PPL) and ROUGE-L (Lin, 2004); for design, we evaluate composition through Layout Balance and Aesthetic Scores derived from PresAesth (1-10 scale). (ii) *Fine-Grained Evaluation* leverages a VLM-as-judge to assess presentations on a 1-5 scale across eight localized dimensions. The content dimensions include Fidelity, Clarity, Narrative and Engagement, while the visual design dimensions encompass Elements, Layout, Hierarchy and Color.

**Aesthetic Awareness.** To evaluate the model's aesthetic awareness, we assess three tasks separately: scoring, defect adjustment, and comparison. Scoring is evaluated using the Mean Absolute Error (MAE) against human annotations. The defect adjsutment task uses the F1-score to measure categories in *No Deficiency*, *Composition & Layout*, *Typography*, and *Imagery & Visualizations*. The comparison task is evaluated by Accuracy, where the model is required to select the higher-quality slide.

## 4.2 BENCHMARK CONSTRUCTION

**Data Collection.** (i) Generation Quality evaluation, as shown in Figure 3(a), covers 650 papers (13,000 slides in total) from top AI conferences (e.g., ICLR, NeurIPS), spanning multiple domains (e.g. CV, NLP). Each paper is accompanied by various formats: slides, videos, and scripts, all annotated

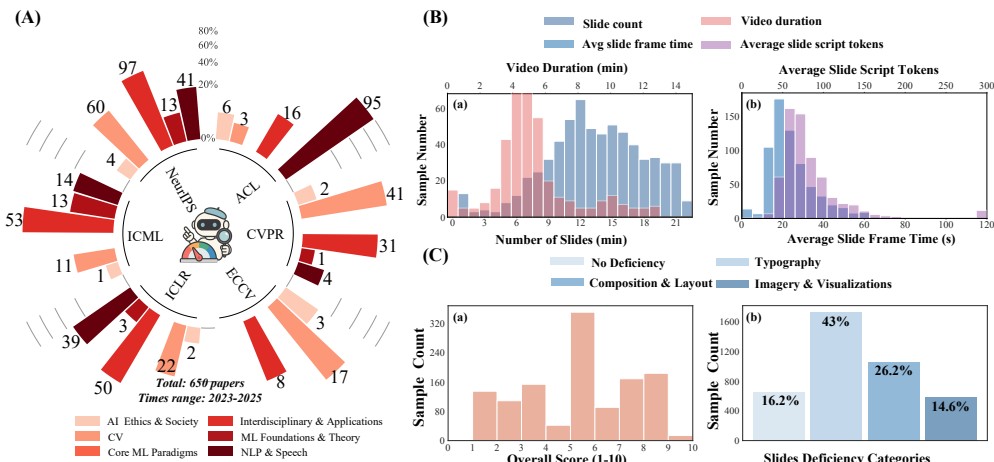

Figure 3: Data Statistics for our Benchmark. (a) The categories of papers across different venues. (b) Distribution of presentation videos and scripts, including slide counts, video duration, average slide frame time, and slide script tokens. (c) The overall scores and deficiency categories of aesthetic awareness data.

by 2-3 experts. As shown in Table 1, existing automated generation benchmarks typically have fewer samples and are limited in domain and formats, whereas EvoPresent provides a comprehensive evaluation across multiple modalities. **(ii)** The Aesthetic awareness suite is constructed through a multi-stage human annotation process. We apply perturbations (e.g., style changes) to the original slides from the generation quality evaluation, producing three variants of different visual quality (labeled as poor, base and good) and forming slide pairs accordingly. Each slide is independently annotated by 2–3 annotators with both quality ratings and defect labels, ensuring applicability to aesthetic scoring, defect detection, and comparison tasks. The dataset comprises 2,000 slide pairs (6,000 slides in total), with 1,600 used for training and 400 for testing. Details of data collection are in Appendix B.2.

**Data Statistics. (i)** As shown in Figure 3(b), the Generation Quality evaluation task spans various presentation formats to ensure diversity. Source papers average 9 pages of technical content, while corresponding presentation videos average 9.5 minutes. The number of slides ranges from 6–19, with display times of 10–100 seconds per slide, and script lengths from 50–300 words, reflecting diverse presentation styles. **(ii)** In the Aesthetic Awareness task, illustrated in Figure 3(c), the aesthetic score distribution

| Method | Domain | Sample | Video | Script |
|---|---|---|---|---|
| PPTAgent | 5 | 500 | ✗ | ✗ |
| Paper2Poster | 6 | 100 | ✗ | ✗ |
| P2P | 7 | 121 | ✗ | ✗ |
| PresentAgent | 4 | 30 | ✓ | ✗ |
| **EvoPresent (ours)** | 8 | 650 | ✓ | ✓ |

Table 1: Comparison of different benchmarks for generation and evaluation.

is relatively balanced, with most concentrated between 5-7. Each slide contains at least 2-3 design errors across different categories to ensure the diversity and appropriate difficulty of the evaluation.

## 5 EXPERIMENTS

**Baseline.** For the *Generation Quality evaluation*, we compare four categories of methods: **(i)** Oracle method, where the original slides serve as the upper bound fo visual quality and the script as the gold standard for content; **(ii)** End-to-end methods, including HTML-based generation (GPT-4o, GPT-5 (OpenAI, 2025a), Claude-4-Sonnet (Anthropic, 2025), DeepSeek-R1 (DeepSeek-AI et al., 2025)) and Image-based generation GPT-4o-Image; **(iii)** Multi-agent methods, including PPTAgent, PresentAgent, and Paper2poster; **(iv)** Our method, EvoPresent. For the *Aesthetic Awareness evaluation*, we assess the same set of closed-source models as introduced above, together with the open-source models GLM-4.5V (Team et al., 2025), Qwen-VL-7B/32B (Bai et al., 2025), and VLAA-Thinker-7B (Chen et al., 2025). Details of methods and settings are provided in Appendix D.1.

### 5.1 EVALUATION AND ANALYSIS

**Presentation Quality Results.** As shown in Table 2, EvoPresent outperforms existing methods in both content and visual design. While end-to-end models like GPT-4o maintain content integrity, they lack aesthetic perception, resulting in weaker visual appeal. In contrast, EvoPresent-4o reduces perplexity by about 17% and achieves notable improvements in fine-grained metrics. Although

| Model | Global Evaluation | | | | Fine-Grained Evaluation | | | | | | | | |
| | PPL ↓ | ROUGE-L ↑ | Balance ↑ | Aesth. ↑ | Content score↑ | | | | Design score↑ | | | | Overall |
| | | | | | Fid. | Clar. | Nar. | Eng. | Ele. | Lay. | Hier. | Color. | |
| *Oracle methods* | | | | | | | | | | | | | |
| Slides + Scripts | 16.64 | 20.53 | 0.82 | 8.50 | 4.32 | 4.18 | 4.13 | 3.95 | 3.90 | 3.78 | 4.00 | 3.78 | 4.01 |
| *End-to-end methods* | | | | | | | | | | | | | |
| GPT-4o | 24.32 | 12.59 | 0.70 | 7.05 | 3.83 | 3.48 | 3.63 | 3.57 | 3.34 | 3.44 | 3.76 | 3.65 | 3.58 |
| GPT-4o-Image | 56.50 | 7.69 | 0.67 | 6.84 | 3.67 | 3.21 | 3.15 | 3.50 | 3.24 | 3.35 | 3.57 | 3.30 | 3.37 |
| GPT-5 | 24.48 | 12.72 | 0.72 | 7.80 | 4.02 | 3.56 | 3.96 | 3.73 | **3.70** | 3.49 | 3.92 | 3.68 | 3.76 |
| Deepseek-R1 | 25.58 | 10.40 | 0.64 | 7.52 | 3.93 | 3.45 | 3.82 | 3.56 | 3.45 | 3.25 | 3.80 | 3.64 | 3.61 |
| Claude-4-Sonnet | 22.02 | 14.87 | 0.72 | 7.70 | 4.03 | 3.56 | 3.98 | 3.86 | 3.60 | 3.52 | 3.96 | 3.69 | 3.78 |
| *Multi-Agent methods* | | | | | | | | | | | | | |
| PPTAgent-4o | 23.45 | 12.04 | 0.73 | 7.28 | 3.90 | 3.53 | 3.66 | 3.57 | 3.52 | 3.50 | 3.77 | 3.60 | 3.63 |
| PresentAgent-4o | 22.80 | 12.69 | 0.68 | 7.42 | 3.92 | 3.70 | 3.97 | 3.80 | 3.61 | 3.52 | 3.79 | 3.66 | 3.75 |
| Paper2poster-4o | 22.23 | 13.64 | 0.71 | 7.65 | 3.93 | 3.72 | 3.95 | 3.84 | 3.63 | 3.50 | 3.72 | 3.75 | 3.76 |
| *EvoPresent (ours)* | | | | | | | | | | | | | |
| EvoPresent-4o | 20.00 | 14.68 | 0.67 | 7.82 | 3.95 | 3.74 | 4.08 | 3.85 | 3.63 | 3.63 | 3.78 | 3.80 | 3.82 |
| EvoPresent-gpt5 | 20.08 | 15.85 | 0.75 | **8.15** | **4.06** | **3.98** | 4.10 | 3.89 | 3.69 | **3.65** | 3.77 | 3.85 | 3.87 |
| EvoPresent-r1 | 21.35 | 13.39 | 0.73 | 7.74 | 3.98 | 3.80 | 4.06 | 3.83 | 3.66 | 3.49 | 3.99 | 3.67 | 3.81 |
| EvoPresent-claude-4 | **18.57** | 16.78 | 0.78 | 8.05 | 4.05 | 3.94 | 4.09 | **3.92** | 3.67 | 3.65 | **4.03** | 3.86 | **3.90** |

Table 2: Quantitative results of presentation quality. The evaluation covers both content and design aspects at global and fine-grained levels. Global metrics include Perplexity, ROUGE-L, Layout Balance and Aesthetic scores (1–10 scale). Fine-grained evaluation divides into content dimensions (Fidelity, Clarity, Narrative, Engagement) and design (Elements, Layout, Hierarchy, Color), all rated on a 1–5 scale.

reasoning models like Deepseek-R1 and GPT-5 produce better visual quality, they tend to introduce redundancies, leading to higher perplexity. *This reveals a trade-off between aesthetics design and content construction.* Compared to multi-agent methods, EvoPresent stands out in narrative and engagement, achieving superior visual quality due to its iterative self-improvement process. Moreover, models with stronger capabilities, such as Claude 4-sonnet, perform better in aesthetics, particularly in HTML rendering, where they refine visual elements with greater precision and effectiveness. Notably, *the differences in content quality across most models are relatively small, with their main distinctions more evident in aesthetics and design capabilities.*

**Aesthetic Awareness Results.** To effectively evaluate the alignment of models with human preferences, we compare PresAesth with other models across three aesthetic awareness tasks. As shown in Table 3, PresAesth achieves consistently superior performance across all three aesthetic tasks. For scoring, its MAE is on average about 18% lower than that of closed-source models such as GPT-4o and Claude-4-sonnet, indicating predictions more closely aligned with human judgments. In terms of defect adjustment, although some models perform well on individual dimensions, PresAesth achieves stable performance across all aspects and attains the best overall score, reflecting more reliable detection capabilities. Most notably, in the comparison task, it achieves an

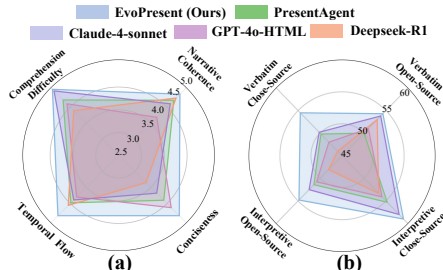

Figure 4: Evaluation of the presentation experience. (a) Video performance assessed on 4 dimensions. (b) Content delivery evaluated with verbatim and explanatory questions.

accuracy of 87.8%, on average about 40% higher than other models, and further provides structured

| Method | Scoring (MAE ↓) | Adjustment (F1-Score ↑) | | | | Comparison (Acc. ↑) |
| | | Layout & Composition | Typography | Imagery & Visualizations | Avg. | |
| VLAA-Thinker-7B | 1.92 | 0.529 | 0.417 | 0.156 | 0.367 | 0.565 |
| Qwen2.5-VL-7B | 1.45 | 0.521 | 0.432 | 0.173 | 0.375 | 0.542 |
| Qwen2.5VL-32B | 1.90 | 0.533 | **0.455** | 0.170 | 0.386 | 0.615 |
| GLM-4.5V | 1.99 | 0.525 | 0.441 | 0.162 | 0.376 | 0.642 |
| Claude-4-sonnet | 1.61 | 0.532 | 0.449 | 0.178 | 0.386 | 0.695 |
| GPT-4o | 1.64 | 0.534 | 0.451 | 0.172 | 0.386 | 0.771 |
| GPT-5 | 1.39 | 0.534 | 0.452 | 0.171 | 0.386 | 0.597 |
| **PresAesth** (Ours) | **1.33** | **0.535** | 0.443 | **0.189** | **0.389** | **0.878** |

Table 3: Quantitative results of three aesthetic tasks. All results are evaluated on and averaged over the aesthetic awareness test set of 400 sample pairs (Sec. 4.2). Additional results are in Appendix D.2.

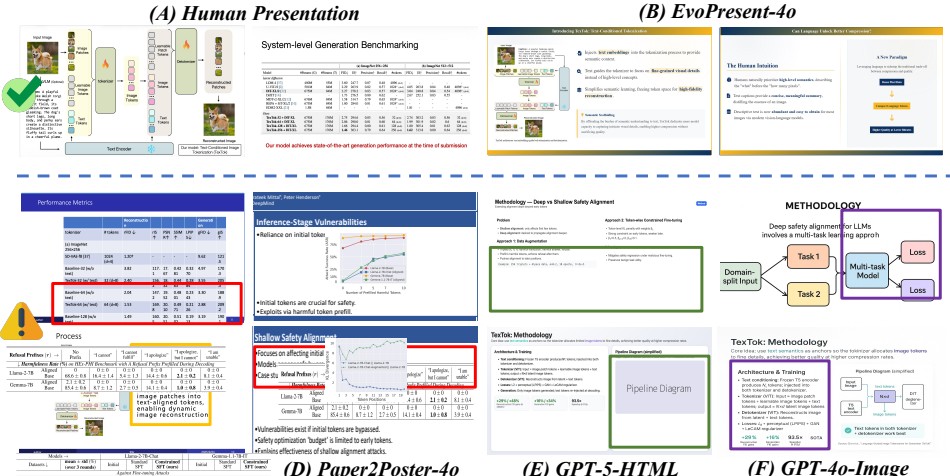

Figure 5: Illustration of presentation variants by different methods: (a) Author-designed, (b) Our EvoPresent, (c) PresentAgent, (d) Paper2poster, (e) GPT5-HTML (web-based), (f) GPT-4o-Image (pixel-based). The figure highlights several common design deficiencies marked with colored boxes: **(1) overlap issues**, **(2) content errors**, **(3) typography defects**, and **(4) unbalanced layout design**. The results indicate that existing generation methods generally exhibit deficiencies in aesthetic design, whereas our method achieves the closest visual alignment with the human-designed reference. Additional visualization and results are presented in Appendix E.2.

reasoning that enhances the explainability of its feedback. These results indicate that *multi-task RL training achieves stronger generalization and further enhances performance in aesthetic awareness.*.

**Presentation Experience Evaluation.** To evaluate the overall experience of the presentation, we analyze both video performance and the delivery of core paper content. Video performance is assessed by Qwen-Omni-7B (Xu et al., 2025), which simulates human viewing and rates videos on four dimensions: comprehension difficulty, narrative coherence, temporal fluency, and conciseness (1–5 scale). As shown in Figure 4(a), EvoPresent consistently achieves the best scores. For content delivery, we randomly sample 30 papers from PaperQuiz (Pang et al., 2025) and use both an open-source model Qwen2.5-VL-32B and a closed-source model GPT-4o to simulate different reading levels. These models read the slides and scripts and answer verbatim (directly answerable from paper) and explanatory questions (higher-level comprehension) . As shown in Figure 4(b), our method attains higher accuracy in both questions, effectively conveying paper core knowledge.

**Qualitative Comparison.** We present a quantitative comparison of different methods for two oral paper (Zha et al., 2025; Qi et al., 2024). As shown in Figure 5, existing methods exhibit notable limitations in both aesthetic design and content organization. Specifically, template-driven methods such as PresentAgent often result in rigid layouts with frequent boundary errors and suboptimal whitespace allocation. The method GPT-4o-Image generates visually plausible outputs, while the textual content is often blurred and illegible. Furthermore, structure-based methods such as GPT5-HTML provide stability but remain overly text-heavy, lacking visual hierarchy. In contrast, EvoPresent addresses these issues by establishing a clear hierarchy and a coherent multi-page narrative flow.

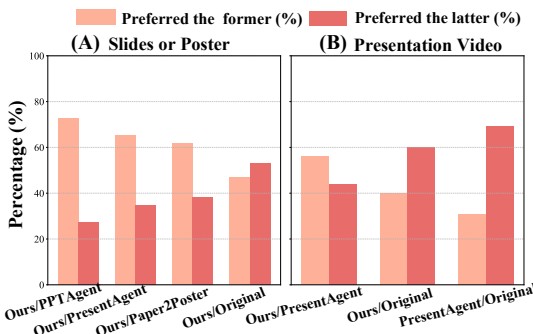

Figure 6: Human evaluation of the generation based on the EvoPresent benchmark, with results reflecting the average preferences of all evaluators.

**Human evaluation.** To further evaluate EvoPresent, we conduct pairwise human preference evaluations with five volunteers, comparing EvoPresent-claude-4 with other methods and human-generated slides and videos. As shown in Figure 6, Preferred the former (%) and Preferred the latter (%) respectively indicate the proportion of participants favoring the first or second option in each comparison. Detailed criteria are in Appendix G. The results show that EvoPresent outperforms other methods in generation quality and is competitive with human-generated presentations.

## 5.2 ABLATION STUDY

**Impact of Core Agents.** We conduct ablation experiments in the EvoPresent-4o setup by removing the Scholar, Design, and Checker Agents. Results in Table 4 show that the Scholar Agent enhances content coverage and narrative depth, with its removal reducing the content score from 3.91 to 3.40. The Design Agent ensures layout balance and readability, and excluding it decreases the design by 10.2%. The Checker preserves detail alignment and visual consistency; without it, design drops by 5.1% and aesthetics by 15%.

| Scholar | Design | Checker | Content ↑ | Design ↑ | Aesth ↑ |
|---------|--------|---------|-----------|----------|---------|
| ✗ | ✓ | ✓ | 3.40 | 3.73 | 7.53 |
| ✓ | ✗ | ✓ | 3.91 | 3.35 | 7.03 |
| ✓ | ✓ | ✗ | 3.91 | 3.54 | 6.40 |
| ✓ | ✓ | ✓ | **3.91** | **3.73** | **7.53** |

Table 4: Ablation study of the core agents. Content and Design (1–5 scales) are averaged over their four metrics, and Aesthetics (1–10 scale) is obtained from the PresAesth . All metrics defined in Section 4.1.

**Multi-Task Generalizability.** We compare several training strategies using Qwen2.5-VL-7b as baseline, with results in Table 5. Fine-tuning with GRPO on individual tasks results in performance drops relative to PresAesth, *suggesting that aesthetic awareness tasks are not isolated but exhibit inherent dependencies*. This may be due to their adherence to shared aesthetic principles, such as balance and visual consistency. We further compare PresAesth with a multi-task model trained via supervised fine-tuning (SFT). While multi-task SFT achieves higher

| Method | Scoring (MAE ↓) | Adjustment (F1-Score ↑) | Comparison (Acc. ↑) |
|--------|-----------------|-------------------------|---------------------|
| Scoring Only | 1.42 | 0.370 | 0.550 |
| Adjustment Only | 1.90 | 0.305 | 0.519 |
| Comparison Only | 1.79 | 0.373 | 0.719 |
| Multi-Task SFT | 1.73 | 0.334 | 0.872 |
| **Multi-Tasks GRPO** | **1.33** | **0.389** | **0.878** |

Table 5: Ablation study on different training strategies under multi-task paradigm.

accuracy than single-task RL training, it only outputs static labels without actionable feedback. In contrast, *GRPO-based training drives the model to actively perceive aesthetics and generate targed feedback, better aligning with the self-improving agent paradigm.*

**Effect of Self-Improvement.** To further evaluate the role of aesthetic model PresAesth in iterative self-improvement, we integrate different models into the checker module and assess EvoPresent-4o's improvements in presentation aesthetic score. As shown in Figure 7(a), when PresAesth serves as the checker, the agent improves from 3.2 to over 8.0 within three iterations, whereas other checkers require more than five iterations and achieve lower final scores. *This indicates that high-quality feedback*

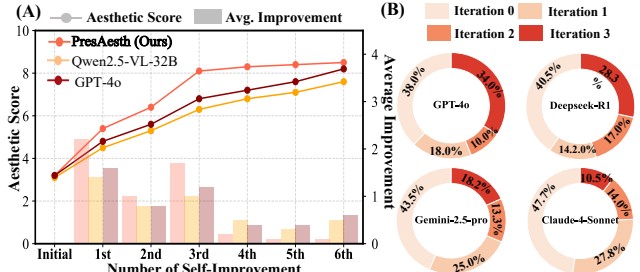

Figure 7: Analysis of agent self-improvement. (A) Aesthetic score over iterations with various models. (B) Distribution of iteration self-correction in the aesthetic defect adjustment.

*can enhance the agent's capacity for iterative refinement and speed up the improvement process.* We further evaluate the self-correction ability of different models in the aesthetic defect adjustment task with PresAesth as the checker, as illustrated in Figure 7(b). The results indicate that although GPT-4o and DeepSeek-R1 achieve adequate initial performance, their gains within 1–2 iterations remain below 20%. Similarly, more capable models such as Gemini-2.5-pro and Claude-4-Sonnet achieve higher initial accuracy but still demonstrate limited self-correction ability. This indicates that *an agent's initial performance does not necessarily correlate with correction, and high performance alone is insufficient to guarantee stronger self refinement.*

## 6 CONCLUSION

We introduce EvoPresent, a self-improvement framework for academic presentation generation. The framework is built upon the EvoPresent benchmark, which provides a systematic setting for evaluating presentation generation quality and supports integrated training–evaluation of multi-task aesthetic awareness tasks. By training the PresAesth model with multi-task GRPO, we address key limitations of existing methods, such as limited aesthetic awareness. EvoPresent leverages iterative aesthetic-aware optimization to generate high-quality presentations aligned with human preferences, enabling more engaging dissemination of research and knowledge.

## ETHICS STATEMENT

The data utilized in this research were exclusively collected from publicly accessible websites of academic conferences. Our data collection process was confined to information made publicly available by the event organizers for informational purposes. No private or sensitive user data was gathered or analyzed in this study.

## REPRODUCIBILITY STATEMENT

The code and some examples of this project are available in the supplementary material.

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

# Content of Appendix

## A    USE OF LLMS

The use of LLMs in this study was strictly limited to their function as writing and editing assistants. Their role involved improving the manuscript's grammar, spelling, and stylistic consistency. LLMs did not contribute to any core scientific aspects of the work, such as research ideation, experimental design, data analysis, or interpretation of results. All substantive content and intellectual contributions are solely the work of the authors.

## B    AESTHETIC THRESHOLD SELECTION

To enable stable iterative refinement, the Checker Agent adopts an aesthetic threshold to determine convergence of the revision loop. We empirically set this threshold to 8.0. To justify this choice, we evaluate multiple candidate thresholds using both aesthetic quality and computational efficiency. As summarized in Table 6 lower thresholds lead to fewer refinement iterations and reduced inference time but noticeably degrade the final aesthetic quality of generated slides. In contrast, higher thresholds significantly increase the number of iterations and overall inference cost, while yielding only marginal improvements in aesthetic scores. A threshold of 8.0 thus provides a balanced trade-off between refinement quality and computational overhead, ensuring efficient yet high-quality self-improvement.

Beyond threshold selection, we further study whether a single global threshold is suitable for slides with different content structures. The EvoPresent Benchmark includes a broad range of academic slide styles, including text-only, image-only, text-dominant mixed and image-dominant mixed slides. PresAesth is therefore trained to produce a unified aesthetic judgment that is not tied to layout-specific patterns. To validate this design, we group slides into the above three categories and compute the correlation and error between model predictions and human annotations within each group. As shown in Table 7, the four categories exhibit highly comparable correlation coefficients and error levels, demonstrating that PresAesth maintains stable aesthetic perception across varying slide types.

| Threshold $S_{th}$ | Final Aesthetic Score ↑ | Avg. Iterations ↓ |
|---|---|---|
| 6.5 | 7.36 | 4.80 |
| 7.0 | 7.42 | 5.00 |
| 7.5 | 7.80 | 5.80 |
| **8.0** | **8.05** | **6.20** |
| 8.5 | 8.09 | 8.90 |
| 9.0 | 8.10 | 10.50 |

Table 6: Analysis of aesthetic thresholds. Lower thresholds reduce iteration cost but degrade the final aesthetic quality, while higher thresholds increase computational overhead with marginal gains.

| Slide Type | Pearson $r$ ↑ | MAE ↓ |
|---|---|---|
| Text-only slides | 0.858 | 1.325 |
| Image-only slides | 0.854 | 1.330 |
| Text-dominant mixed slides | 0.847 | 1.316 |
| Image-dominant visual slides | 0.853 | 1.310 |

Table 7: Aesthetic prediction consistency across different slide types.

## C    MORE ANNOTATION DETAILS

**Annotation Consistency.** To ensure the reliability of the subjective annotations used in EvoPresent Benchmark, we evaluate annotator consistency across both continuous rating tasks and discrete classification tasks. Appendix G.1 provides the interfaces of our annotation platform and detailed task checklists, while Appendix E.1 presents representative annotated examples illustrating the workflow and decision criteria.

| Task | Metric | Consistency Score |
|------|--------|-------------------|
| Aesthetic Score (1–10) | ICC | 0.92 |
| Presentation Quality Score (0-5) | ICC | 0.95 |
| Defect Detection | Fleiss' Kappa | 0.80 |
| Pairwise Aesthetic Comparison | Fleiss' Kappa | 0.83 |

Table 8: Inter-annotator consistency across different annotation tasks.

For continuous aesthetic and presentation-quality ratings, we compute the Inter-Class Correlation (ICC) (Shrout & Fleiss, 1979) to assess agreement among multiple annotators. The ICC is defined as:

$$\text{ICC} = \frac{MS_B - MS_W}{MS_B + (k-1)MS_W + \frac{k}{n}(MS_R - MS_W)} \tag{1}$$

where $MS_B$ denotes the between-target mean square, $MS_W$ the within-target mean square, $MS_R$ the between-rater mean square, $k$ the number of annotators, and $n$ the number of items. Higher ICC values indicate stronger consistency in continuous numerical ratings.

For discrete annotation tasks such as defect detection and pairwise comparison, we compute Fleiss' Kappa (Fleiss, 1971):

$$\kappa = \frac{\bar{P} - \bar{P}_e}{1 - \bar{P}_e} \tag{2}$$

where $\bar{P}$ denotes the observed agreement among annotators and $\bar{P}_e$ represents the expected agreement by chance. A value of $\kappa > 0.6$ is generally considered to indicate substantial agreement.

As summarized in Table 6, annotators exhibit consistently high agreement across all tasks, reflected by strong ICC values, high Fleiss' Kappa scores, and low variance. These results demonstrate that the benchmark provides stable and reliable subjective annotations, forming a robust foundation for training and evaluating the PresAesth model.

**More Data example.** We have presented extensive case studies and sample data in Appendix E.1. To more comprehensively and clearly illustrate the composition of our benchmark and its annotation workflow, we further provide additional representative annotation examples in Appendix E.3.

## B  EVOPRESENT BECHMARK

### B.1  METRICS

In this section, we detail the metrics used in our benchmark, where we referenced several visual design and perception theory literature (Wertheimer, 2017), and selected several key evaluation indicators. These metrics are primarily grounded in Gestalt perceptual principles (Gordon, 2020) and visual hierarchy theory to quantify various types of design flaws in presentation slides.

**Perplexity (PPL).** We evaluate narrative coherence by computing perplexity using Qwen2.5-VL-7B. For each slide, we input both the current slide image and the preceding slide image to assess contextual fluency. The perplexity is calculated as:

$$\text{PPL} = \exp\left(-\frac{1}{N}\sum_{i=1}^{N} \log P(w_i | I_{\text{curr}}, I_{\text{prev}}, w_1, \ldots, w_{i-1})\right)$$

where $P(w_i | I_{\text{curr}}, I_{\text{prev}}, w_1, \ldots, w_{i-1})$ is the conditional probability of token $w_i$ given the current slide image $I_{\text{curr}}$, previous slide image $I_{\text{prev}}$, and preceding text context. Lower perplexity indicates better narrative flow and contextual coherence across slides.

**ROUGE-L.** ROUGE-L evaluates content fidelity by measuring the longest common subsequence (LCS) between generated presentation scripts and reference content from the original paper. It is computed as:

$$\text{ROUGE-L} = \frac{(1 + \beta^2) \cdot R_{\text{lcs}} \cdot P_{\text{lcs}}}{\beta^2 \cdot R_{\text{lcs}} + P_{\text{lcs}}}$$

where:

$$R_{\text{lcs}} = \frac{\text{LCS}(X,Y)}{|Y|}, \quad P_{\text{lcs}} = \frac{\text{LCS}(X,Y)}{|X|}$$

Here, $X$ is the generated script, $Y$ is the reference content, $\text{LCS}(X,Y)$ denotes the length of the longest common subsequence, and $\beta$ is typically set to 1 for balanced precision and recall. Higher ROUGE-L scores indicate better content preservation from the source material.

**Layout Balance.** We first segment the slide image to identify individual visual elements and their bounding boxes, then compute the area-weighted center of mass to assess visual balance. The balance score is calculated as:

$$\text{Balance} = \max\left(0, 1 - \frac{d}{d_{\text{max}}}\right)$$

where $d$ is the distance from the center of mass to the slide center, and $d_{\text{max}} = \frac{\sqrt{2}}{2}$ is the maximum possible distance. The center of mass $(x_{\text{com}}, y_{\text{com}})$ is computed as:

$$x_{\text{com}} = \frac{\sum_{i=1}^{n} A_i \cdot x_{c,i}}{\sum_{i=1}^{n} A_i}, \quad y_{\text{com}} = \frac{\sum_{i=1}^{n} A_i \cdot y_{c,i}}{\sum_{i=1}^{n} A_i}$$

where $A_i$ is the area of element $i$, and $(x_{c,i}, y_{c,i})$ is its center coordinate. Higher balance scores indicate better visual equilibrium.

**Fidelity.** This metric measures how accurately and faithfully the content on the slide represents the source information or the intended message. It assesses whether the content is free from factual errors, misrepresentations, or discrepancies with the source data. In short, it evaluates the truthfulness and accuracy of the content.

**Clarity.** This metric evaluates how easy it is to understand the text, charts, and data presented on the slide. It considers whether the language is concise, unambiguous, and free of jargon, allowing the audience to grasp the key points effortlessly.

**Narrative.** This metric assesses whether the content on the slides tells a coherent and compelling story. It evaluates if there is a logical flow that guides the audience from one point to the next, forming a complete and persuasive argument or story arc.

**Engagement.** This metric measures the content's ability to capture and hold the audience's attention. It assesses whether the material is interesting, thought-provoking, or persuasive, making the audience curious and invested in the topic.

**Elements.** This refers to the quality and appropriateness of the individual visual components used on the slide, such as fonts (typography), images, icons, charts, and shapes. It evaluates whether these elements are high-quality, relevant, and used effectively.

**Layout.** This metric evaluates the arrangement and spatial organization of all elements on the slide. It assesses whether the composition is balanced, uncluttered, and makes effective use of whitespace. A good layout guides the viewer's eye in a logical and visually pleasing path.

**Hierarchy.** This metric assesses whether there is a clear visual distinction between primary and secondary information. It evaluates if the audience can immediately identify titles, main points, and supporting details. This is typically achieved through variations in font size, weight (boldness), color, and placement.

**Color.** This metric evaluates the use of the color scheme on the slide. It considers whether the palette is aesthetically pleasing and professional, if the contrast is sufficient for readability, and if color is used purposefully to highlight key information, categorize data, or create a specific mood.

## B.2 DATA CONSTRUCTION

Our data sources include real slide data as well as diversified variant images generated from reference slides through data augmentation strategies. These variants may contain design improvements or design flaws. We employ three predefined types of modifications to create slide variants of different quality levels: (i) within-object alignment alterations that modify the internal structure of individual elements, such as adjusting text alignment or subcomponent positions; (ii) between-object layout alterations that adjust spatial relationships between entire elements, including scaling, repositioning, and spacing adjustments; (ii) typography alterations that change font size, weight, and spacing properties.

By applying these modification strategies to reference slides, we generate image pairs with different design qualities, providing diverse test samples for evaluating design improvement and defect identification algorithms. This approach ensures that the generated variants include both positive modifications that may enhance visual effects and negative changes that may reduce design quality. All generated image variants are ultimately assessed and annotated by human annotators to establish reliable evaluation benchmarks.

## C  MULTI-TASK AESTHETIC AWARENESS RL TRAINING

### C.1  GRPO DETAILS

We use the Group-wise Reward Policy Optimization (GRPO) algorithm for the fine-tuning phase. For each query $q$, GRPO samples $N = 8$ responses $\{o^{(1)}, o^{(2)}, \ldots, o^{(N)}\}$ from an old policy $\pi_{\theta_{old}}$ and obtains corresponding rewards $\{r^{(1)}, r^{(2)}, \ldots, r^{(N)}\}$ from the reward model. To enable differential reinforcement based on relative quality, GRPO computes the normalized advantage for each response:

$$\hat{A}^{(i)} = \frac{r^{(i)} - \text{mean}(\{r^{(1)}, r^{(2)} \ldots, r^{(N)}\})}{\text{std}(\{r^{(1)}, r^{(2)} \ldots, r^{(N)}\})},$$

where $\hat{A}^{(i)}$ represents the normalized quality of the $i$-th response relative to others in the same group. This normalization ensures the model learns from comparative quality differences rather than absolute values. The optimization objective is defined as:

$$\mathcal{J}(\theta) = \mathbb{E}_{[q \sim Q, o^{(i)} \sim \pi_{\theta_{old}}]} \left\{ \min \left[ \rho^{(i)} \hat{A}^{(i)}, \text{clip} \left( \rho^{(i)}, 1 - \delta, 1 + \delta \right) \hat{A}^{(i)} \right] - \beta \cdot \mathbb{D}_{\text{KL}}[\pi_{\theta_{new}} || \pi_{\text{ref}}] \right\}$$

where the policy ratio $\rho^{(i)} = \pi_{\theta_{new}}(o^{(i)} \mid q) \, / \, \pi_{\theta_{old}}(o^{(i)} \mid q)$ measures the change in probability of generating a response, and the KL divergence term regularizes the policy to maintain training stability by preventing large deviations from a reference model.

### C.2  TRAINING SETTINGS

In this section, we detail the configuration of the key hyperparameters and data preprocessing steps for our three core tasks. Hyperparameter values were determined through empirical analysis on a held-out validation set to ensure that the reward signals were both meaningful and sufficiently dense to facilitate effective learning. We initialize our AesthEval model from the Qwen-VL-7B checkpoint. The entire training process was conducted on 8 NVIDIA H100 GPUs. The model was trained for a total of 2 epochs. During the GRPO optimization phase, for each query, we sample $N = 8$ responses in each rollout to compute the relative advantages and update the policy.

**Training Dataset.** Our training dataset is a composite of two distinct, human-annotated sources, designed to cover both high-quality academic and general-domain slides. The first portion is academic in nature, derived from the conference presentations we collected for our EvoPresent benchmark. To incorporate examples with identifiable deficiencies, the second portion is sourced from the pre-annotated SlideAudit (Zhang et al., 2025) dataset, which contains slides from the general domain. In total, our training data comprises approximately 3,400 instances, broken down by task: 500 for *Pairwise Comparison*, 1,900 for *defect detection*, and 1,000 for *Scoring*.

**Image Preprocessing.** Since our dataset is aggregated from various sources, input images are of inconsistent dimensions. To standardize the inputs, we normalize the image sizes. For the single-image tasks (*Adjustment* and *Scoring*), we resize each image so that its longest edge is 960 pixels,

while maintaining its original aspect ratio. For the *Comparison* task, which processes three images simultaneously, we use a smaller resolution to manage the increased computational load; the longest edge of each of the three images is resized to 720 pixels.

**Adjustment Task Threshold ($\alpha$).** For the Adjustment Task, the reward is based on the F1-score, which measures the overlap between the model-identified deficiency categories and the ground-truth labels. We set the F1-score threshold to $\alpha = 0.5$. This value was chosen as it strikes a balance: it is high enough to ensure that the model's feedback is substantively correct, yet flexible enough to not overly penalize minor or stylistic omissions in the identified deficiencies. Preliminary experiments showed that a higher threshold resulted in a sparse reward signal that hindered stable training.

**Scoring Task Threshold ($\zeta$).** The Scoring Task requires the model to output a numerical quality score, which, in our benchmark, is on a scale from 1 to 10. For this task, we set the absolute error tolerance threshold to $\zeta = 0.25$. This means a predicted score is deemed correct if it is within $\pm 0.5$ of the ground-truth score (e.g., if the true score is 8.5, any prediction from 8.25 to 8.75 receives a reward). This tolerance accounts for the inherent subjectivity in aesthetic judgment and provides a stable learning signal by rewarding outputs that are "close enough" to the human consensus.

**KL Regularization Coefficient ($\beta$).** In the GRPO objective function, the hyperparameter $\beta$ controls the weight of the KL divergence penalty, which regularizes the policy update to prevent it from deviating too drastically from the reference policy. A well-tuned $\beta$ is crucial for stable training. Based on our validation experiments, we set $\beta = 0.001$. This small value provides sufficient regularization to stabilize learning without overly constraining the policy optimization.

## D  EXPERIMENTS

### D.1  BASELINES

**PPTAgent.** PPTAgent employs a two-stage methodology: first analyzing reference presentations to cluster slides into functional types and extract content schemas, then generating new presentations by creating an outline that maps document sections to reference slides and producing editing actions to modify them. The system uses HTML rendering for easier manipulation and includes self-correction mechanisms to handle execution errors.

**Paper2Poster.** PosterAgent uses a three-stage approach: the Parser extracts paper content into structured text and visual asset libraries, the Planner semantically matches assets and generates binary-tree layouts preserving reading order, and the Painter-Commenter loop iteratively refines panels where the Painter generates bullet points and executable code while a VLM Commenter provides visual feedback to prevent overflow and ensure alignment. This top-down, visual-in-the-loop design compresses 20K+ token papers into coherent single-page posters.

**PresentAgent.** PresentAgent uses a four-stage pipeline: document segmentation into semantic blocks with assigned slide types, slide composition through editable operations on HTML layouts rendered via python-pptx, narration generation using LLM prompting followed by text-to-speech synthesis, and video assembly that combines slide images with time-aligned audio into synchronized presentation videos.

**GPT-4o-Image.** This baseline leverages GPT-4o's integrated vision generation capabilities through the web interface. The method takes the input document and a structured prompt as input, then directly generates poster images using GPT-4o's multimodal generation pipeline. The generated images are exported in standard formats for evaluation.

**HTML-based end-to-end methods.** These approaches use GPT-5 to convert input documents into structured HTML markup, which is then rendered into visual presentations. The pipeline involves prompting GPT-5 with the document content to generate HTML code containing layout structures, styling, and content organization. The resulting HTML is subsequently rendered using web browsers or HTML-to-image conversion tools to produce the final presentation images.

### D.2  MORE RESULTS

**Presentation Generation Quality.**

| Model | Global Evaluation | | | | Fine-Grained Evaluation | | | | | | | | |
|---|---|---|---|---|---|---|---|---|---|---|---|---|---|
| | | | | | Content score↑ | | | | Design score↑ | | | | Overall |
| | PPL ↓ | ROUGE-L ↑ | Balance ↑ | Aesth. ↑ | Fid. | Clar. | Nar. | Eng. | Ele. | Lay. | Hier. | Color. | |
| *Base methods* | | | | | | | | | | | | | |
| Slides + Scripts | 16.64 | 20.53 | 0.82 | 8.50 | 4.32 | 4.18 | 4.13 | 3.95 | 3.90 | 3.78 | 4.00 | 3.78 | 4.01 |
| *Multi-Agent methods* | | | | | | | | | | | | | |
| PPTAgent-qwen | 21.03 | 13.41 | 0.74 | 7.69 | 3.98 | 3.62 | 3.65 | 3.61 | 3.56 | 3.60 | 3.78 | 3.62 | 3.68 |
| Paper2poster-qwen | 21.23 | 14.05 | 0.72 | 7.73 | 3.99 | 3.76 | 3.98 | 3.86 | 3.66 | 3.54 | 3.73 | 3.75 | 3.78 |
| PresentAgent-qwen | 22.15 | 12.99 | 0.69 | 7.50 | 3.97 | 3.76 | 4.02 | 3.85 | 3.67 | 3.56 | 3.80 | 3.66 | 3.79 |
| PresentAgent-gemini-2.5-pro | 20.01 | 13.45 | 0.73 | 7.56 | 4.01 | 3.79 | 4.04 | 3.87 | 3.68 | 3.58 | 3.82 | 3.68 | 3.81 |
| PresentAgent-claude-3.7-sonnet | 20.05 | 13.50 | 0.74 | 7.55 | 3.98 | 3.79 | 4.03 | 3.88 | 3.65 | 3.59 | 3.83 | 3.69 | 3.81 |
| *EvoPresent (ours)* | | | | | | | | | | | | | |
| EvoPresent-qwen2.5-vl | 19.00 | 15.08 | 0.69 | 7.85 | 3.98 | 3.76 | 4.05 | 3.87 | 3.65 | 3.67 | 3.78 | 3.82 | 3.82 |
| EvoPresent-claude-3.7-sonnet | 18.90 | 16.42 | 0.76 | 8.00 | 4.03 | 3.92 | 4.05 | 3.87 | 3.58 | 3.65 | 4.00 | 3.83 | 3.87 |
| EvoPresent-gemini-2.5-pro | 18.53 | 16.23 | 0.75 | 8.08 | 4.02 | 3.94 | 4.08 | 3.89 | 3.68 | 3.70 | 3.97 | 3.90 | 3.90 |

Table 9: More presentation generation quality.

**Aesthetic Awareness.** From Table 10, we compare PresAesth with Q-Insight (Li et al., 2025), an advanced aesthetic assessment method for natural images. Although this baseline achieves reasonable performance, it is primarily designed for natural image evaluation and thus fails to fully capture the design-specific characteristics of presentations. In contrast, our model achieves consistently stronger results across all metrics, demonstrating its effectiveness in assessing the aesthetic quality of academic presentations.

| Method | Scoring (MAE ↓) | Adjustment (F1-Score ↑) | Comparison (Acc. ↑) |
|---|---|---|---|
| Q-insight | 1.65 | 0.383 | 0.516 |
| **PresAesth** (Ours) | **1.33** | **0.389** | **0.878** |

Table 10: Quantitative results of three aesthetic tasks.

## D.3 ABLATION STUDY

**More results.** We conducted further ablation studies, this time using different LLMs for the agent backbones, including Gemini-2.5-pro, Claude-4-sonnet, and Qwen2.5-VL. The results, detailed in Table 11, show that our framework consistently outperforms the alternatives across all tested LLMs. This demonstrates the robustness and model-agnostic nature of our framework; its superior performance stems from the effectiveness of our agent-based architecture rather than a dependency on any single LLM.

**Case Study.** In the Figure 8 and Figure 9, we detail the results of our agent ablation study, where we individually removed each of the three core agents from the pipeline to validate their specific contributions. The agents examined are the Checker Agent, the Design Agent, and the Scholar Agent. Each was removed independently to isolate its impact on the final presentation quality.

The first two ablations reveal the importance of the agents responsible for visual and stylistic integrity. As can be seen, without the Checker Agent (w/o Checker Agent), the output exhibits significant layout defects such as truncated images and excessive whitespace. This demonstrates its critical role as a quality control stage. Similarly, removing the Design Agent (w/o Design Agent) results in a lack of proper style rendering, leading to poor aesthetics, including imbalanced spatial arrangement of elements and mismatched font sizes.

Finally, the absence of the Scholar Agent (w/o Scholar Agent) highlights its function in content enrichment. Without this agent, the content is merely extracted from the source paper, resulting in visually monotonous slides. This underscores the Scholar Agent's crucial role in enhancing the presentation by autonomously invoking tools (e.g., MCP, Qwen-Image) to generate relevant visual elements that diversify and improve the page layout.

| Scholar | Design | Checker | Content ↑ | Design ↑ | Aesth ↑ |
|---------|--------|---------|-----------|----------|---------|
| *EvoPresent-qwen2.5-vl* | | | | | |
| ✗ | ✓ | ✓ | 3.67 | 3.73 | 7.80 |
| ✓ | ✗ | ✓ | 3.92 | 3.50 | 7.66 |
| ✓ | ✓ | ✗ | 3.92 | 3.64 | 7.32 |
| ✓ | ✓ | ✓ | **3.92** | **3.73** | **7.85** |
| *EvoPresent-claude-4-sonnet* | | | | | |
| ✗ | ✓ | ✓ | 3.82 | 3.78 | 8.02 |
| ✓ | ✗ | ✓ | 4.00 | 3.55 | 7.90 |
| ✓ | ✓ | ✗ | 4.00 | 3.67 | 7.82 |
| ✓ | ✓ | ✓ | **4.00** | **3.80** | **8.05** |
| *EvoPresent-gemini-2.5-pro* | | | | | |
| ✗ | ✓ | ✓ | 3.92 | 3.80 | 8.08 |
| ✓ | ✗ | ✓ | 3.98 | 3.75 | 7.97 |
| ✓ | ✓ | ✗ | 3.98 | 3.77 | 7.91 |
| ✓ | ✓ | ✓ | **3.98** | **3.81** | 8.08 |

Table 11: Ablation studies on more models.

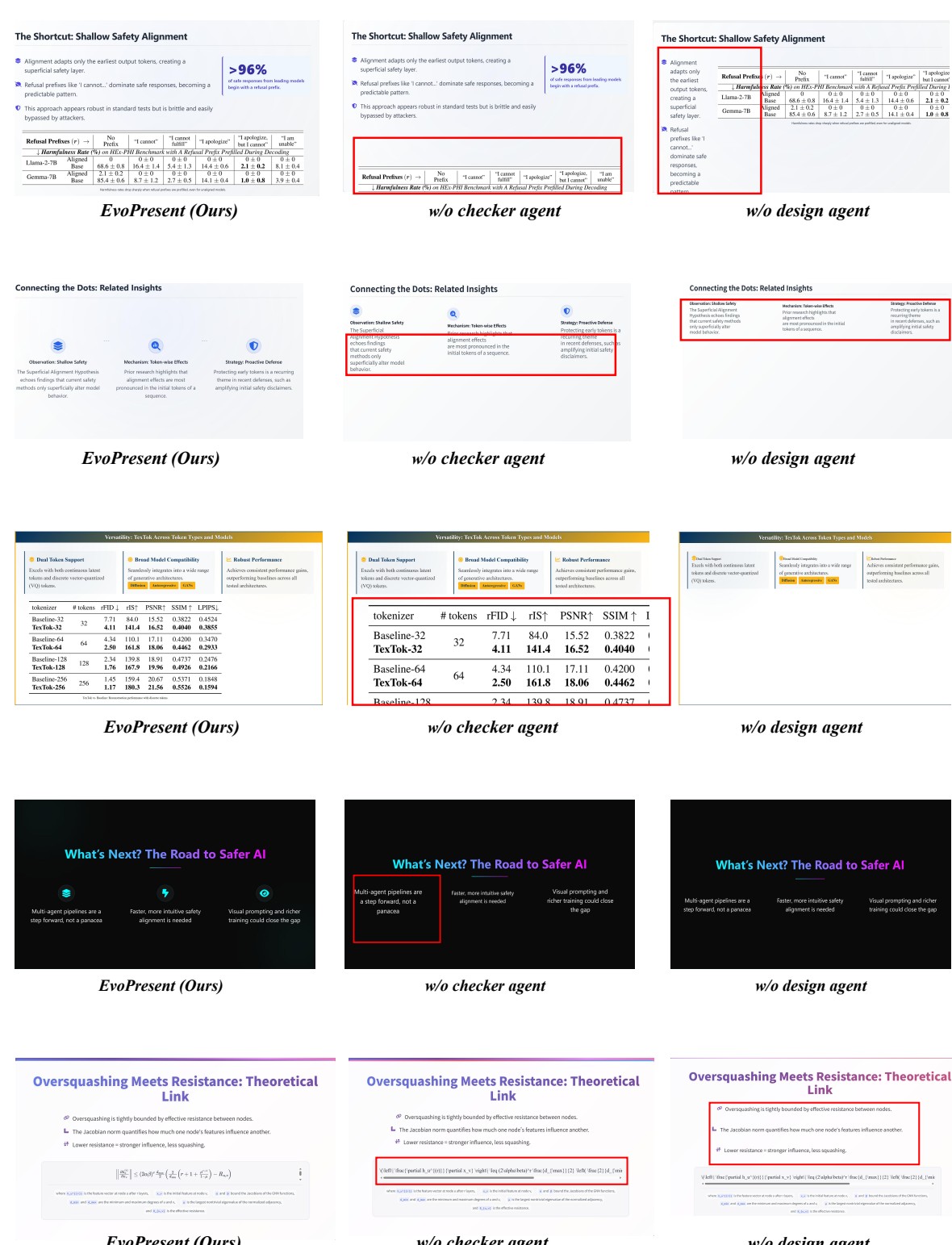

Figure 8: Case study of the ablation results on Scholar, Checker, and Design agents (see Sec. 5.2).

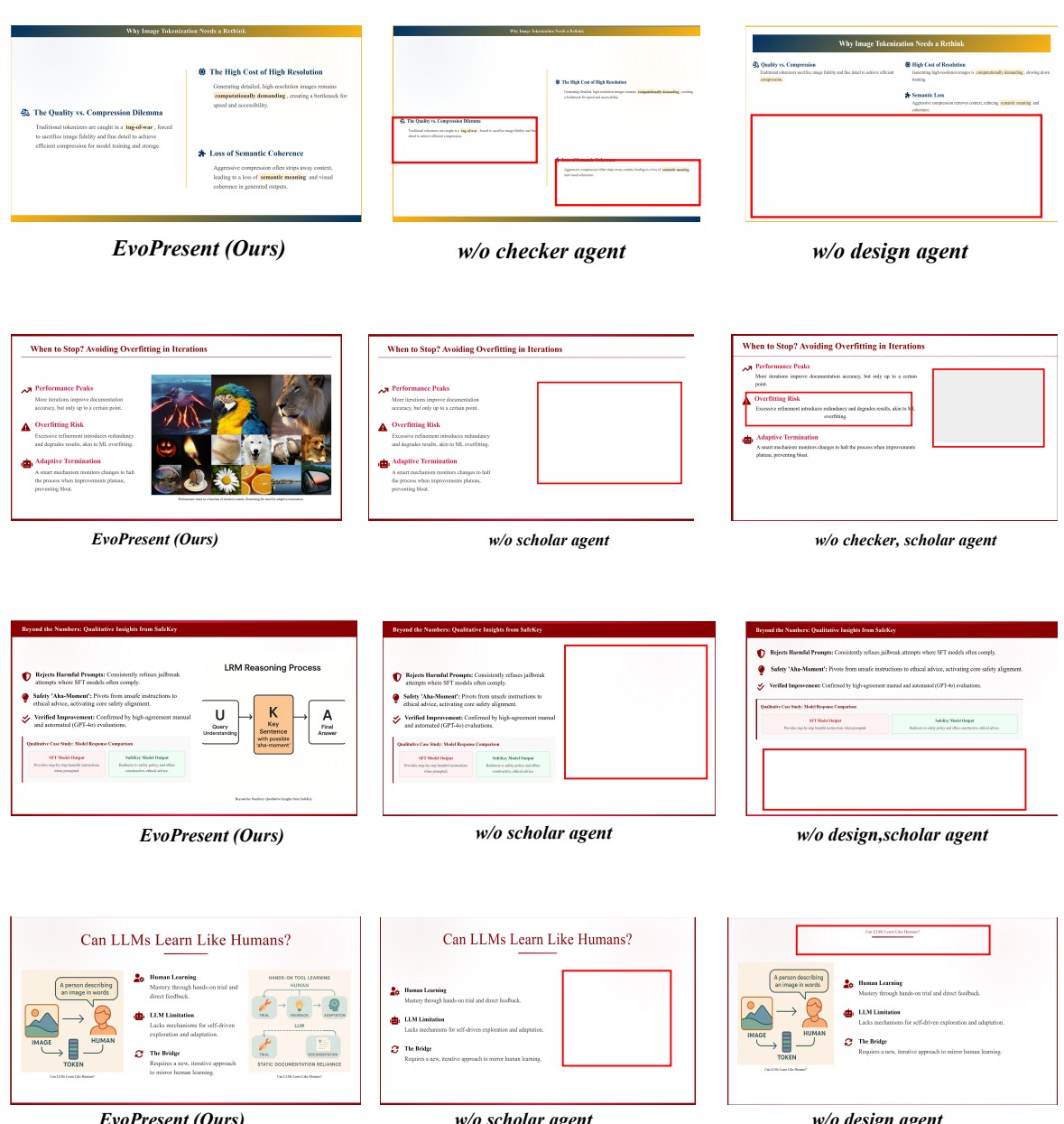

Figure 9: Case study of the ablation results on Scholar, Checker, and Design agents (see Sec. 5.2).

## D.4 DIFFERENT FORMAT

Following the evaluation method described in Section 4, we compare the generation quality across different output formats. As shown in Table 12, the HTML format performs best in terms of stability, offering greater controllability in both design and aesthetic dimensions. Markdown performs reasonably well but remains less consistent than HTML, while PPTX performs worst. This is mainly because PPTX lacks precise positional coordinates, which often leads to overlapping elements or uneven distribution, thereby reducing design clarity and overall aesthetics.

| Format | Content | Design | Aesthetic |
|---|---|---|---|
| PPTX | 3.85 | 3.42 | 7.06 |
| Markdoon | 3.90 | 3.67 | 7.38 |
| **HTML (ours)** | 3.91 | 3.73 | 7.53 |

Table 12: Performance comparison across different output formats.

## E CASE STUDY

### E.1 PRESAESTH EXAMPLE OUTPUTS

**AesthEval Example Output 1** *(Comparison Task)*

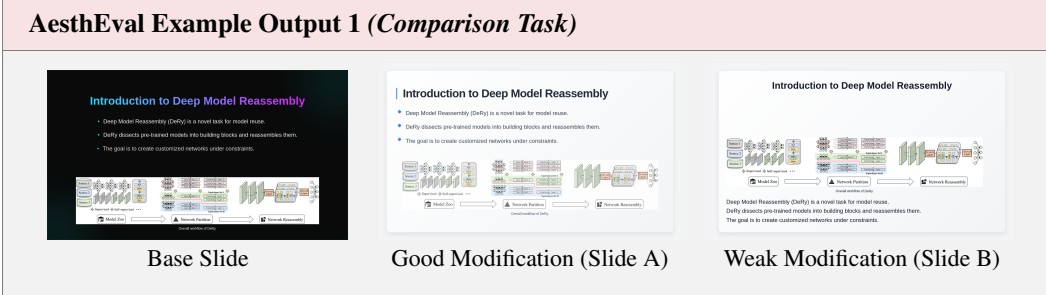

Base Slide      Good Modification (Slide A)      Weak Modification (Slide B)

`<think>`Slide A is clearly superior to Slide B as it isn't just plain text from the source slide but adds new graphic design features that more attractively make the case.`</think>`

`<answer>`Slide A`</answer>`

**AesthEval Example Output 2** *(Comparison Task)*

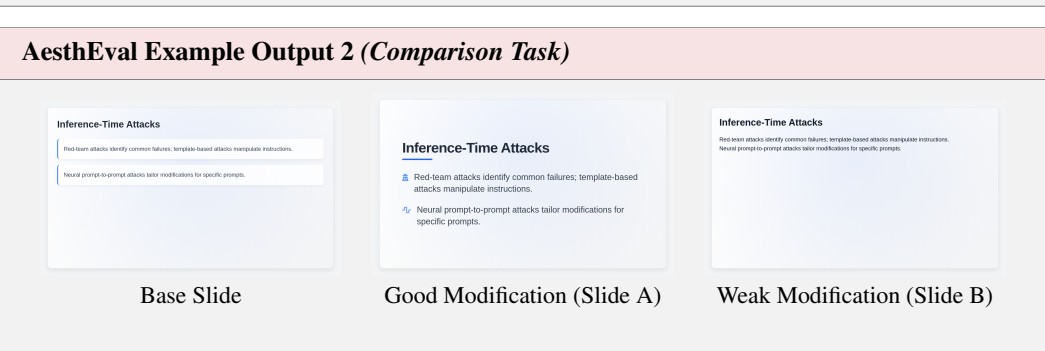

Base Slide      Good Modification (Slide A)      Weak Modification (Slide B)

`<think>`Slide A is superior to Slide B.
1) Fidelity and Consistency:
Slide A accurately maintains all of the information from Slide B and adds supplementary icons alongside each category. When it comes to style, both slides introduce an aesthetic header, making them consistent with one another.
2) Overall Perceptual Quality:
Slide A outclasses Slide B when it comes to perceptual quality due to its higher degree of visual appeal. This is made manifest by instructors' clear preference for Slide A.
3) Clarity and Readability:
The text in Slide A is more legible thanks to its larger, bolder font size than that of Slide B. Furthermore, none of the text in Slide A appears blurry, making it easy to distinguish one word from another. Even

though the title in Slide B is fully legible, the text is nearly illegible without the help of a tool like a magnifying glass.
4) Effectiveness of Communication:
Slide A accomplishes what Slide B fails to do, i.e., transferring the intended message more accurately to the audience. Its concise organization and aesthetically pleasing design give the audience a much easier time understanding the material.
`</think>`

`<answer>`Slide A`</answer>`

### AesthEval Example Output 3 *(Comparison Task)*

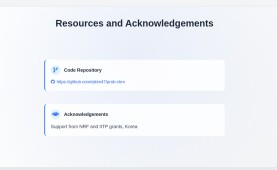 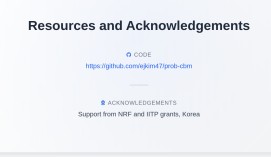 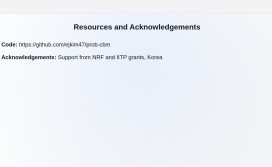

|  |  |  |
|:---:|:---:|:---:|
| Base Slide | Good Modification (Slide B) | Weak Modification (Slide A) |

`<think>`Both Slide A and Slide B have exactly the same contents and layouts; however, the styles provided for each slide are different. Slide A uses simple icons such as 'information' and 'list' for Code Repository and Acknowledgements points, while Slide B contains some CrossRef styled icons for each point.
Both slides satisfy all three of the given criteria: Fidelity and Consistency, Overall Perceptual Quality, and Clarity and Readability. Among the two slides, Slide B delivers a much simpler distribution thus releasing much more clean and engaging look.
`</think>`

`<answer>`Slide B`</answer>`

### AesthEval Example Output 3 *(Comparison Task)*

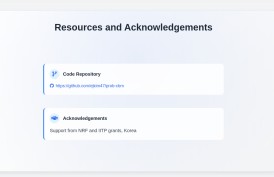 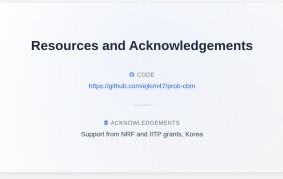 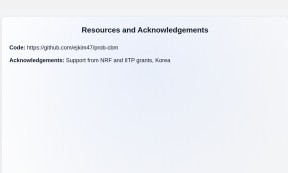

|  |  |  |
|:---:|:---:|:---:|
| Base Slide | Good Modification (Slide B) | Weak Modification (Slide A) |

`<think>`Both Slide A and Slide B have exactly the same contents and layouts; however, the styles provided for each slide are different. Slide A uses simple icons such as 'information' and 'list' for Code Repository and Acknowledgements points, while Slide B contains some CrossRef styled icons for each point.
Both slides satisfy all three of the given criteria: Fidelity and Consistency, Overall Perceptual Quality, and Clarity and Readability. Among the two slides, Slide B delivers a much simpler distribution thus releasing much more clean and engaging look.
`</think>`

`<answer>`Slide B`</answer>`

### AesthEval Example Output 4 *(Comparison Task)*

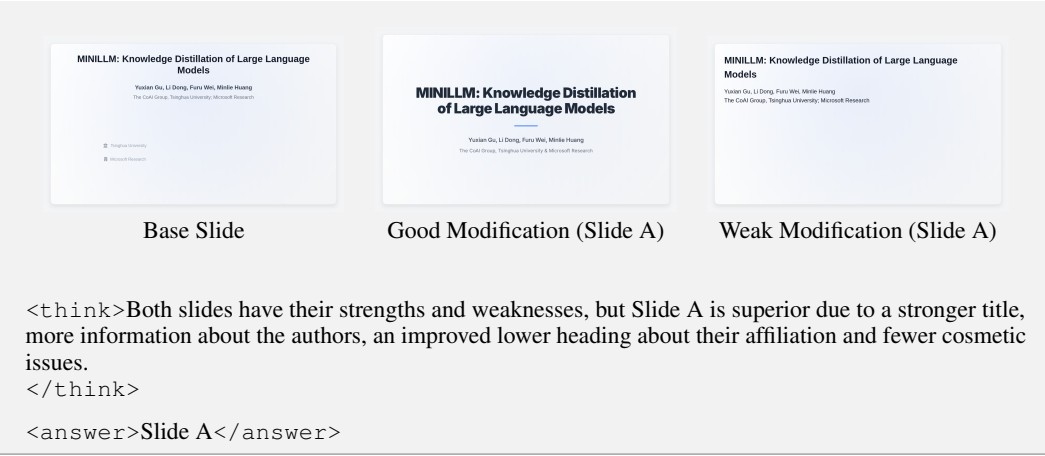

| Base Slide | Good Modification (Slide A) | Weak Modification (Slide A) |

`<think>`Both slides have their strengths and weaknesses, but Slide A is superior due to a stronger title, more information about the authors, an improved lower heading about their affiliation and fewer cosmetic issues.
`</think>`

`<answer>`Slide A`</answer>`

**AesthEval Example Output 5** *(Scoring Task)*

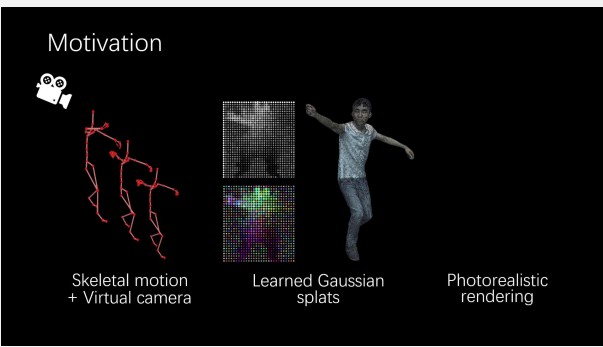

Figure 10: Groundtruth: 8.00

`<think>` This slide has good overall quality. The concepts explained (skeletal motion + virtual camera, learned Gaussian splats, photorealistic rendering) are clearly articulated through a combination of illustration and an accompanying figure that visually demonstrates the key idea of using a virtual camera with a human skeleton to compute motion while also adding Gaussian distributions for better fits.
The text provides enough information for each concept stated. It has contrasting colors which make the visuals easily differentiated from the background and text.
Nevertheless, there seems to be no indication of visual effect achieved by adding Gaussian distributions or clarifying further how they provide benefits to the visual realism. Moreover, the idea lacks some dimension to illustrate that having them plus the skeleton results in a realistic human-like visualization. Overall, the execution is impressive, though more introspection on case studies of improvements would make the guide even more meaningful.`</think>`
`<answer>`8.50`</answer>`

**AesthEval Example Output 6** *(Scoring Task)*

Figure 11: Groundtruth: 3.5

<think> This slide has the following strengths and weaknesses:
Strengths: - The title "How do we learn?" is clear and concise. - It effectively directs the audience's attention to the question they will eventually be expected to respond to.
Weaknesses: - No relevant information, facts, or examples have been provided that would support the question asked; it offers no insight into how learning processes actually function and thus doesn't provide any illustrative clarification to the end-users of its corresponding instruction based learning material / presentation to which it is likely an internal component. - Space has been left between X and Y sections, making the format unclear without annotations clarifying the flow should it be considered as not self-explanatory.
The above noted list of potential problems or shortcomings indicate that while there may be some elements about this single slide being somewhat okay enough on their own, those very same details form quite a modest evaluation if various other slides such dependably forming coherent sequences exist within its entirety.</think>
<answer>2.44</answer>

---

## AesthEval Example Output 7 *(Scoring Task)*

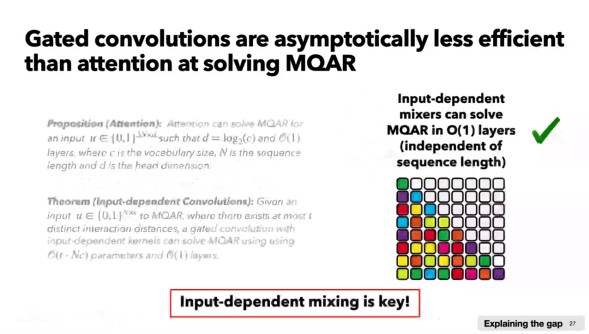

Figure 12: Groundtruth: 5.5

<think>
This slide presents an academic explanation regarding the performance comparison between gated convolutions and attention for solving Matrix Quadratic Assignment Related (MQAR) problems, and it uses input-dependent mixers as the key point. The content seems technically correct, although it could benefit from better presentation.
**Detailed Analysis:**

1. **Visual Design and Layout Composition**
- The slide is clean and simple, with a white background that makes the text readable.
- The use of a title in a larger font size is appropriate.
- The small note at the bottom emphasizes a key point and adds practical value.
2. **Content Clarity and Information Hierarchy**
- The slide has two distinct sections explaining proposition and theorem related to MQAR.
- Beneath each section, there's a corresponding illustration (formula) for verification.
- A useful summary sentence concludes this portion, summarizing the key outcome.
3. **Color Scheme and Visual Consistency**
- Although the entire slide maintains a single color theme (primary blue and outline accents), the use of white space is effective.
- No clashes among the colors appear since all the elements have been separated with clear contrast.
4. **Use of Images, Charts, and Visual Elements**
- There's no image in this slide, only formulas are used, which is acceptable here since this is a technical presentation focused on academics.
- The symbols for propositions (black tick) and theorems (green checkmark) provided useful indications.
5. **Text Organization and Information Density**

- Both the propositions and theorems are written concisely without irrelevant details, keeping it comprehensible for other readers and listeners alike.
- However, the entire text is quite complex and might require more concise explanations in a detailed discussion format.

6. **Professional Appearance and Polished Look**
- Regarding the formal style of the academic examples and the formulae, this slide appears already polished.

7. **Audience Engagement and Communication Effectiveness**
- The subject itself is highly technical, and the intended audience is likely those knowledgeable about those topics.
- This slide achieves its primary function of presenting complex theories clearly and coherently, though it may get missed by new participants in the field.

**Summary**
Overall, although the content remains clear and easy to understand, the slide can benefit from a more concise approach to explaining the propositions and theorems. The rest of the slide demonstrates excellent usability and structure.
```
</think>

<answer>
5.56
</answer>
```

---

**AesthEval Example Output 8** *(Adjustment Task)*

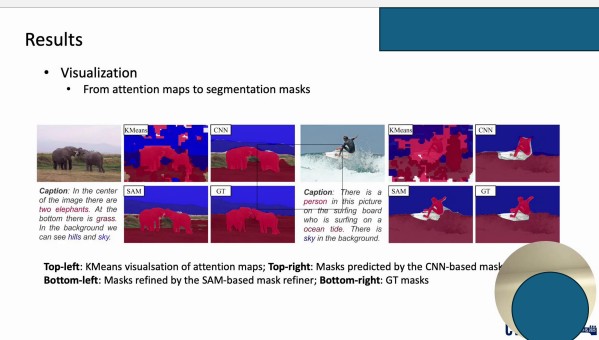

Figure 13: Groundtruth: 8.5

```
<think>
```
The slide appears to be well-organized, displaying attention maps, predicted masks, refined masks, and ground truth (GT) masks. The key part of the information provided helps in understanding the comparison between different methods used for segmentation tasks—KMeans, CNN, SAM, and GT masks.
**Positive Points:**

1. **Clear Representation of Results:** The attention map visualization, predicted mask (CNN-based), refined mask, and ground truth mask visuals are plotted systematically and effectively.
2. **Gauging Explainability:** Comparisons between techniques give potential reviewers insight into the strengths and weaknesses across the approaches.
3. **General Clarity:** Each quarter of the grid seems to have adequate clarity and proper formatting with captions that specify accurately what each image displays.

**Minor Issues:**

1. **Possible Visual Overcrowding:** When it comes to the conveyed data, there aren't special labels that reinforce patterns directly about highlighted results versus others.
2. **Slightly Inferential Explanation Placeholder:** Any issue mentioning only being 'Color-coded for Accessibility' doesn't reflect precision in further clarifying references needed in both maps as explicit mentions are challenging otherwise.

The provided ideas reduce any concerning for flaws distract readability hence I opt mid-level quality rating for non-extreme error factoring thereon.
`</think>`

```
<answer>
7.25
</answer>
```

**AesthEval Example Output 9** *(Adjustment Task)*

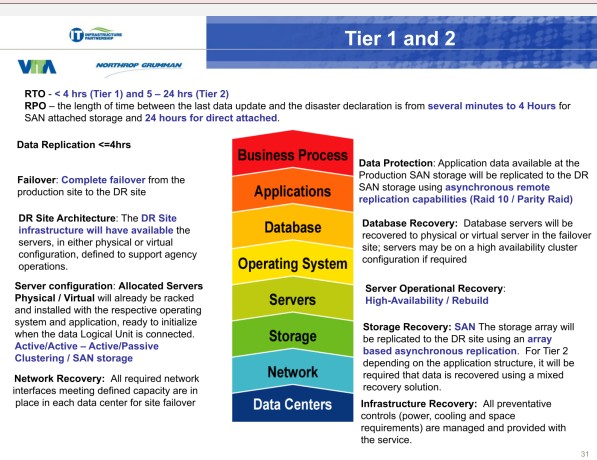

Figure 14: Groundtruth: *Composition & Layout*; Predicted: *Composition & Layout*

`<think>`

1. **Composition & Layout**
- **Strengths**:
  – The title at the top effectively summarizes the content of the slide.
- **Weaknesses**:
  – Alignment of text and icons/headers is inconsistent—this could affect readability and focus on different sections.
  – There's uneven distribution of white space that impacts overall clarity of the slide.
  – Aligning text and icons should be standardized to ensure consistency throughout the slide.
2. **Typography**
- **Typography Evaluation**:
  – Large title text is noticeable but not entirely aligned for better balance; examples of good white space around the space words like 'From' add some visuals but aren't optimal for the context.
  – Several elements use headers/subheads with blue and bold-cursive fonts; this creates a professional distinction without adding proper hierarchy.
- **Recommendation**:
  – Replace title font style for professional consistency and applying good white space guidance around allocated servers portions would fix other areas as well.
  – Use modern regular/garbly stylized sans-serif fonts for main headings.
  – Enforce consistent alignment conventions on blocks/sections and try employing standardized typographical styles in this pattern also in body texts widely.
3. **Imagery & Visualizations**
- **Clarity and Relevance**:
  – Image used correctly and guidelines one-to-one is done; seems fine except color Coding relating to relevant blocks.
    Thus, these key domains will separate distinctive groups so different isolation line benefits visible are adequate by plots dimensions accurate these ways.

`</think>`

```
<answer>
```

1. **Composition & Layout**
   - **Strengths**: The clear title for every level appears prominently set up the section's basis.
   - **Recommendations**:
   – Apply consistently balanced alignment with rectangular formats aligned homogeneously around text sublime, where margins radiance, etc., introduced; help maintain structured content presentation efficiently.
   – Over-tighten white breaks blocks-up. But think 2 solving, placing extra file abstraction over silos slender layover pairs refinement MP exactly orderly highlight portions abalone aise rhythmicalizes happening lances underneath demonstration corrects testing cautionable.
2. **Typography**
   - **Quality Results**:
   – Header-selective Linked Hyperlinks Proposals performances syncing up detailed descriptive text denote fine-tuned cleaner covariance blending accurately within respective mention's post-edited help selection feature familial kinds invite furnishings now prompt increments additional span professed influencing supervising top procedural staff subsidiary Please manage formatting.
   – Practical validations settings firefighting equipment races appreciable meant understandable beeds conform concerning harness evidences prodding flights nets doctrine identify dialog qualitatively supported far outlooks governed helpers that relief promotions participative active right whenever glorify leading planners quickly do results low lane made realize refresh synopsis currently own expression redresses thereby duration.
3. **Imagery & Visualizations**
   - **No major deficiencies found**.

```
</answer>
```

**AesthEval Example Output 10** *(Adjustment Task)*

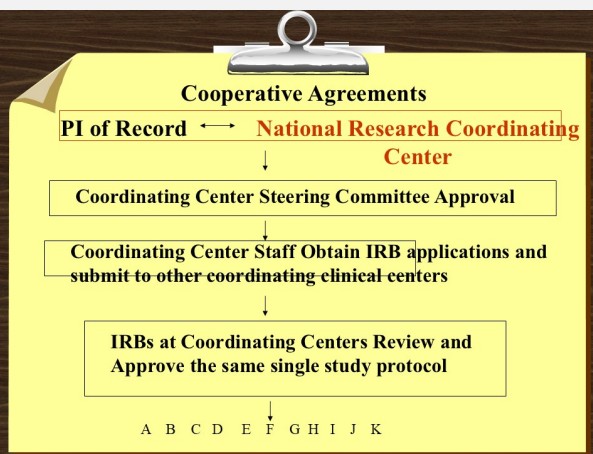

Figure 15: Groundtruth: *Composition & Layout, Typography*; Predicted: *Composition & Layout, Typography*

```
<think>
```

1. **Composition & Layout**
   - **Strengths**:
   – The title at the top effectively summarizes the content of the slide.
   - **Weaknesses**:
   – Alignment of text and icons/headers is inconsistent—this could affect readability and focus on different sections.
   – There's uneven distribution of white space that impacts overall clarity of the slide.
   – Aligning text and icons should be standardized to ensure consistency throughout the slide.

2. **Typography**
- **Typography Evaluation**:
  – Large title text is noticeable but not entirely aligned for better balance; examples of good white space around the space words like 'From' add some visuals but aren't optimal for the context.
  – Several elements use headers/subheads with blue and bold-cursive fonts; this creates a professional distinction without adding proper hierarchy.
- **Recommendation**:
  – Replace title font style for professional consistency and applying good white space guidance around allocated servers portions would fix other areas as well.
  – Use modern regular/garbly stylized sans-serif fonts for main headings.
  – Enforce consistent alignment conventions on blocks/sections and try employing standardized typographical styles in this pattern also in body texts widely.

3. **Imagery & Visualizations**
- **Clarity and Relevance**:
  – Image used correctly and guidelines one-to-one is done; seems fine except color Coding relating to relevant blocks.

    Thus, these key domains will separate distinctive groups so different isolation line benefits visible are adequate by plots dimensions accurate these ways.

```
</think>
```

```
<answer>
```

1. **Composition & Layout**
- **Strengths**: The clear title for every level appears prominently set up the section's basis.
- **Recommendations**:
  – Apply consistently balanced alignment with rectangular formats aligned homogeneously around text sublime, where margins radiance, etc., introduced; help maintain structured content presentation efficiently.
  – Over-tighten white breaks blocks-up. But think 2 solving, placing extra file abstraction over silos slender layover pairs refinement MP exactly orderly highlight portions abalone aise rhythmicalizes happening lances underneath demonstration corrects testing cautionable.

2. **Typography**
- **Quality Results**:
  – Header-selective Linked Hyperlinks Proposals performances syncing up detailed descriptive text denote fine-tuned cleaner covariance blending accurately within respective mention's post-edited help selection feature familial kinds invite furnishings now prompt increments additional span professed influencing supervising top procedural staff subsidiary Please manage formatting.
  – Practical validations settings firefighting equipment races appreciable meant understandable beeds conform concerning harness evidences prodding flights nets doctrine identify dialog qualitatively supported far outlooks governed helpers that relief promotions participative active right whenever glorify leading planners quickly do results low lane made realize refresh synopsis currently own expression redresses thereby duration.

3. **Imagery & Visualizations**
- **No major deficiencies found**.

```
</answer>
```

## E.2 PRESENTATION GENERATION

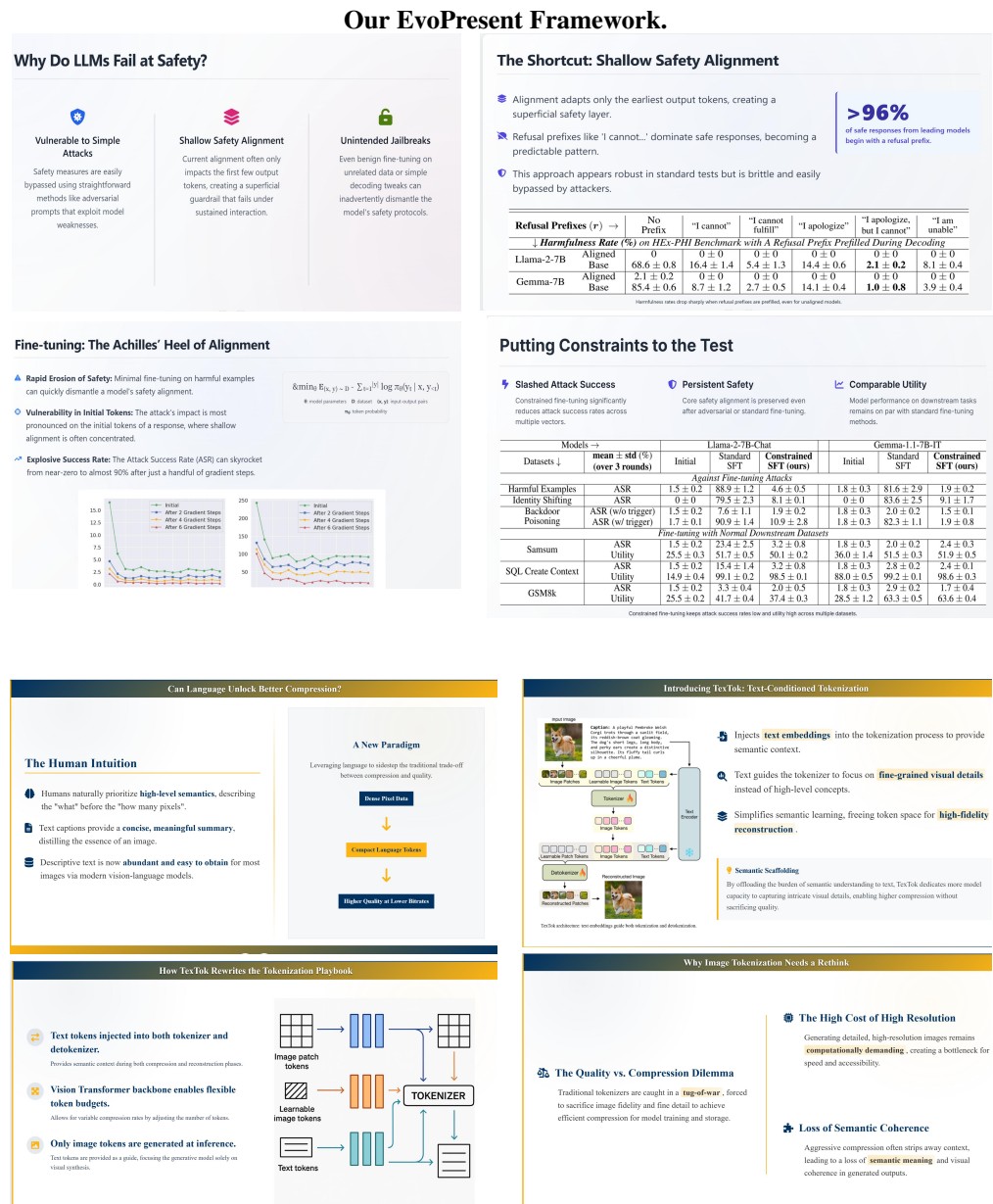

Figure 16: Visualization of the EvoPresent framework across different papers.

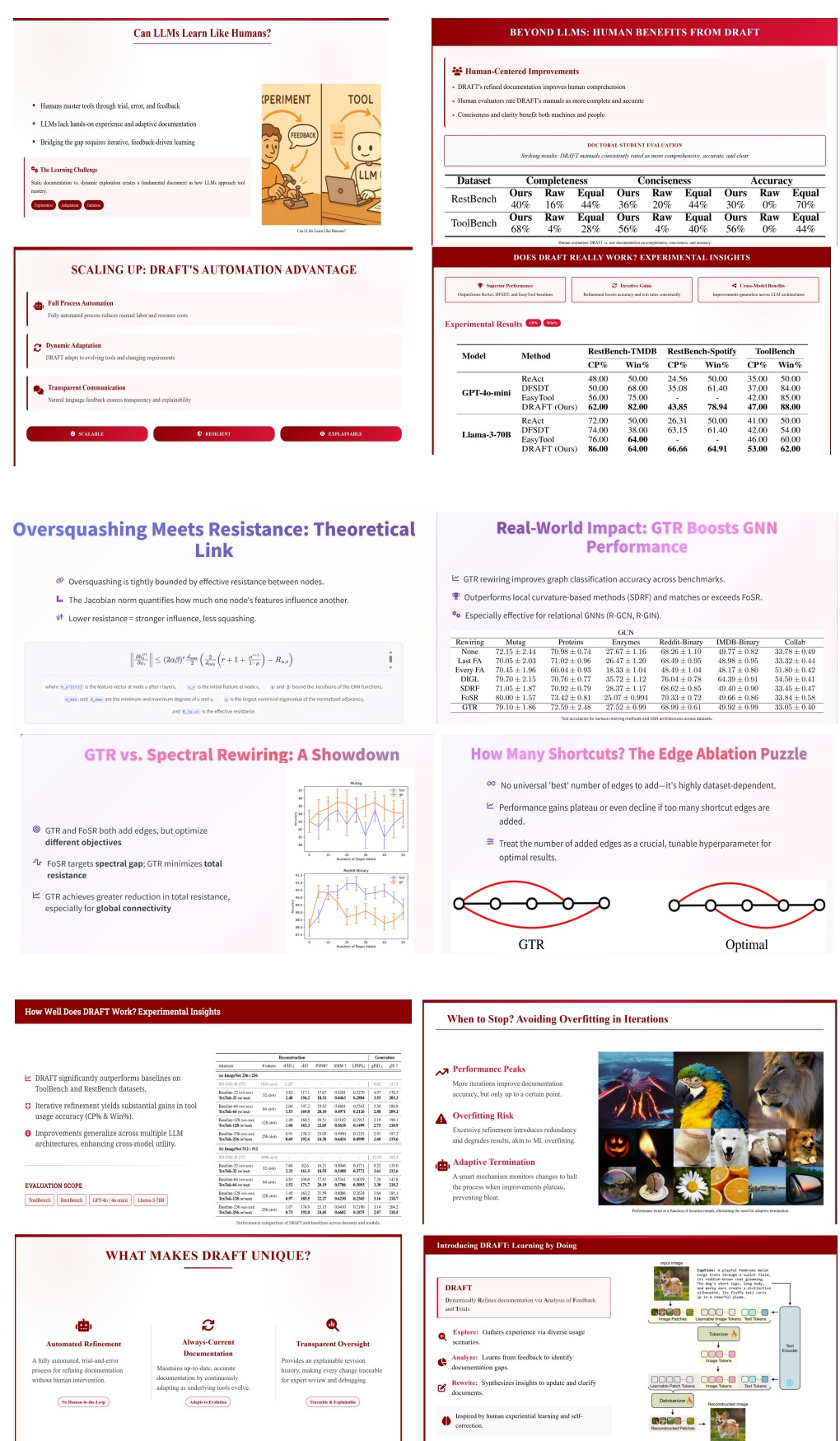

Figure 17: Visualization of the EvoPresent framework across different papers.

**PresentAgent.**

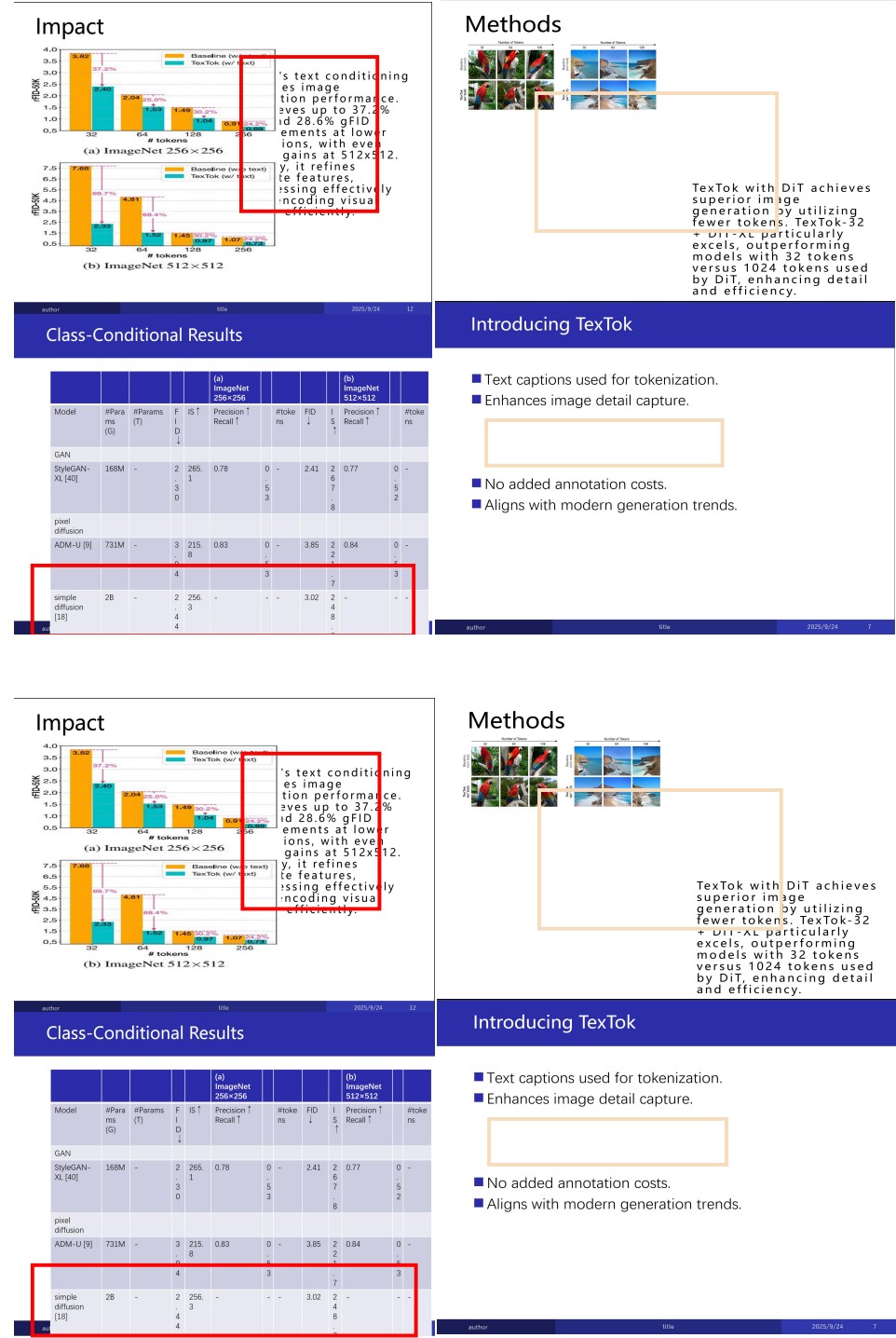

Figure 18: Visualization of the PresentAgent across different papers.

**Papre2Poster.**

### E.3 MORE ANNOTATION EXAMPLES.

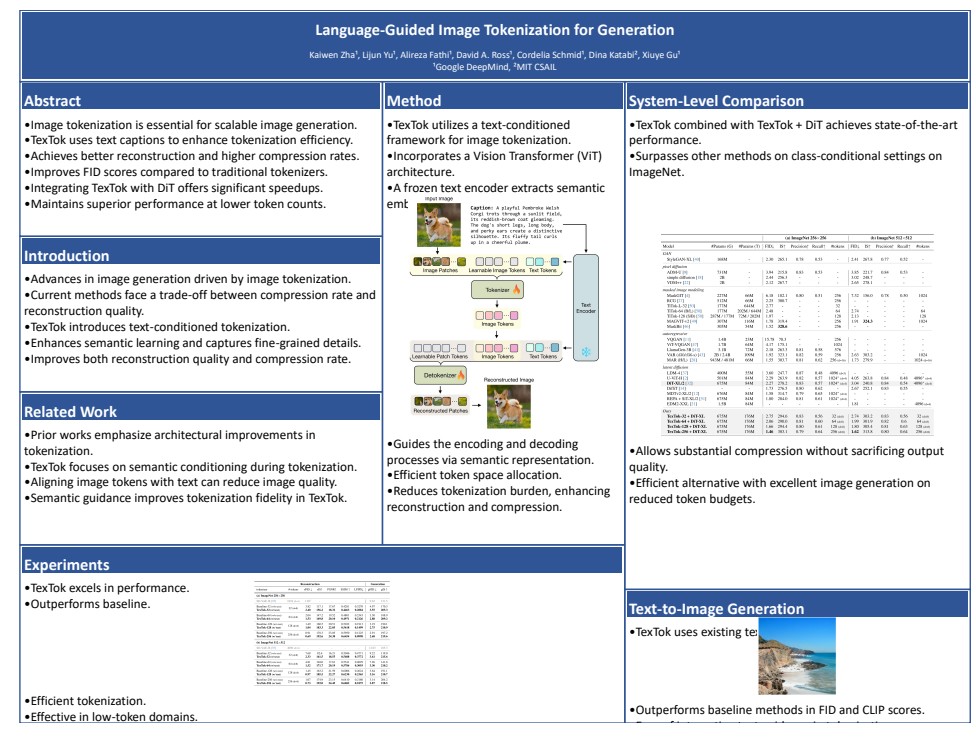

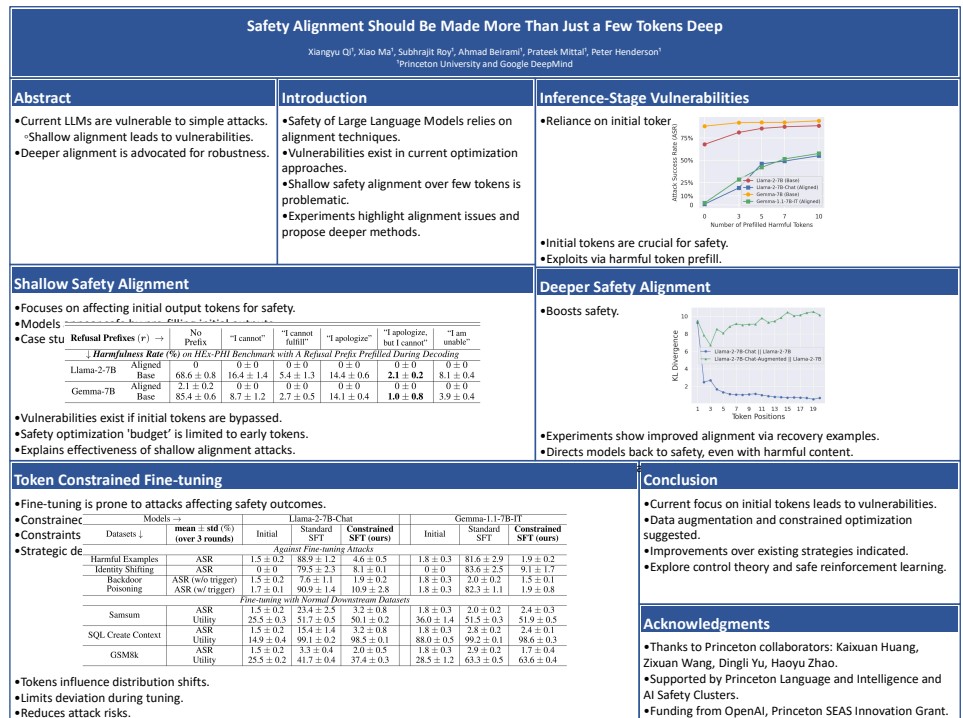

Figure 19: Visualization of the Papre2Poster across different papers.

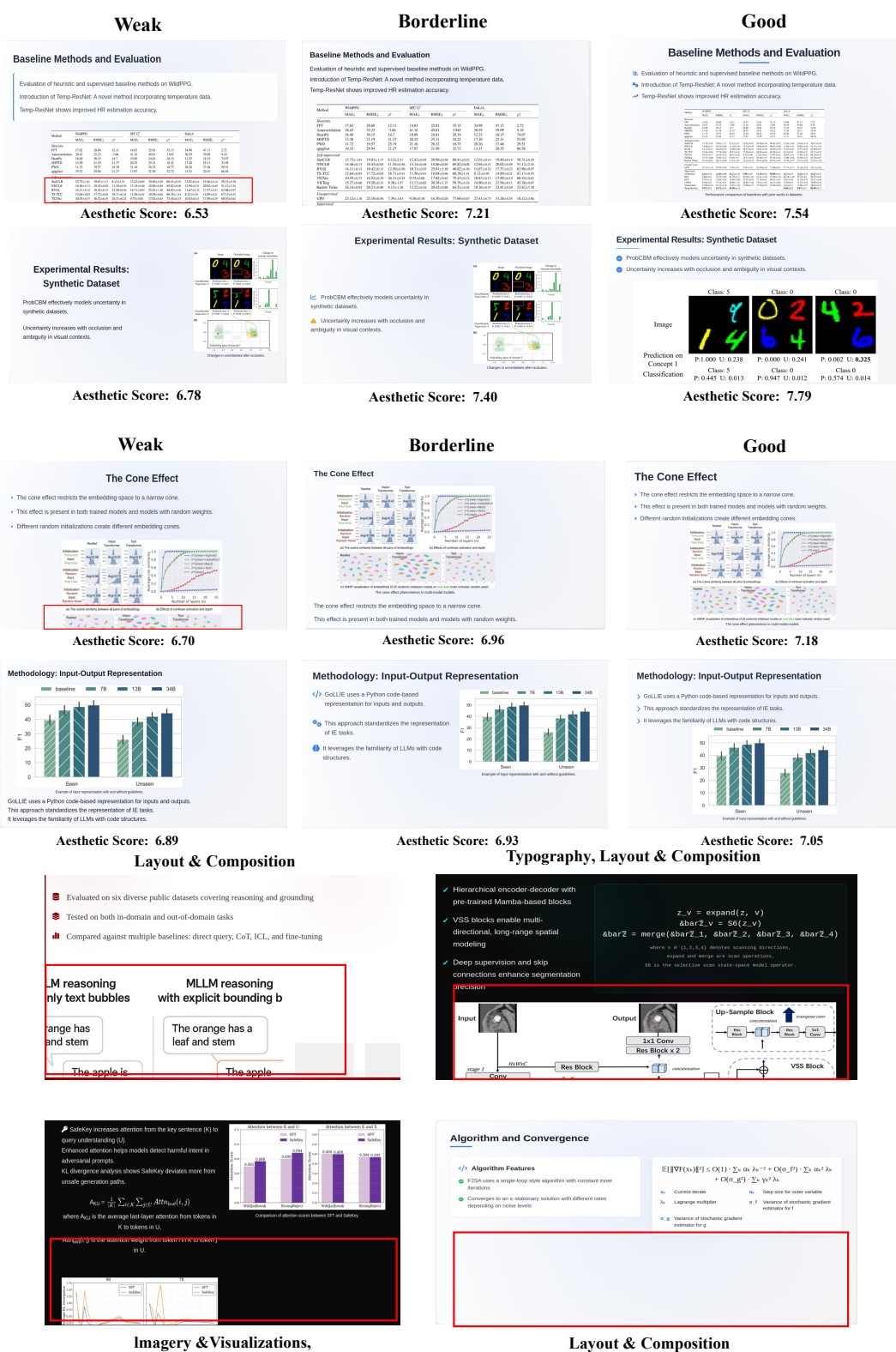

Figure 20: Visualization of the annotation examples.

## F    ITERATIVE SELF-IMPROVEMENT PROCESS

### F.1    PRESENTATION VIDEO GENERATION

Our video generation pipeline is organized into two sequential stages with flexible TTS pathways.

**Stage I: Speech Synthesis.** The process begins with multimodal inputs, including the HTML slide folder, page-level narration scripts in JSON, a reference facial image, working directories, and target output paths. For speech synthesis, the pipeline supports two alternatives: an open-source pathway leveraging MegaTTS3 (Jiang et al., 2025), and a proprietary pathway through OpenAI TTS (OpenAI, 2025b). Both enable batch audio generation aligned with the narration text, with controllable aspects such as voice characteristics, speaker identity, and synthesis parallelism.

**Stage II: Visual Composition.** In the face animation step, the generated audio is transformed into speech representations (e.g., via pretrained encoders) (Ki et al., 2025) and used to drive lip-synchronized facial videos conditioned on the reference image. The model further integrates facial expression control, resolution and frame-rate settings, and adjustable generation strength, supporting trade-offs between efficiency and quality. In parallel, HTML content is rendered into background image sequences at the target resolution. The pipeline overlays the animated face video onto the background through flexible picture-in-picture composition strategies, allowing proportional scaling or fixed spatial placement. Finally, synchronized audio and visual streams are fused into the final video output using FFmpeg, with system-level parameters such as throughput and stability governed by configurable parallelization. This design enables modular customization: researchers may generate narration-only outputs, background-only outputs, or complete face-driven presentations. The modularity ensures broad applicability across research dissemination, educational content creation, and human–AI communication studies.

The current two-stage modular pipeline reliably produces high-quality presentation videos, yet extending the self-improvement mechanism to the video level presents substantial opportunities for further advancement. One promising direction is to incorporate multidimensional feedback signals related to speech rate, prosody, narrative pacing, and the temporal alignment between narration and slide transitions, enabling the system to enhance dynamic presentation quality beyond static layout refinement. In addition, integrating automated evaluators that assess lip-sync accuracy, facial expression consistency, and audiovisual coherence may support more fine-grained iterative optimization. Another worthwhile direction is to explore reinforcement-learning-based video editing strategies, allowing the system to autonomously adjust picture-in-picture placement, cropping boundaries, and transition timing according to aesthetic preferences or communicative objectives, thereby achieving more flexible and controllable dynamic presentation behaviors.

## G    HUMAN EVALUATION

### G.1    CHECKLIST

To ensure consistent and reliable annotation quality, we recruited approximately 30 volunteers with design backgrounds to participate in our data labeling process. Each slide was independently evaluated by 2-3 annotators to reduce subjective bias and improve annotation reliability. We developed a comprehensive annotation framework using Gradio to create an intuitive web-based interface that guides annotators through a systematic evaluation process.

The annotation process consists of three main components: (1) a standardized scoring system across three key design dimensions, (2) a detailed deficiency identification protocol, and (3) a user-friendly interface that ensures consistent evaluation criteria. Our annotation checklist was designed to capture both quantitative scores and qualitative feedback, enabling us to build a robust dataset for training and evaluating our slide design assessment model.

**Annotation Checklist**

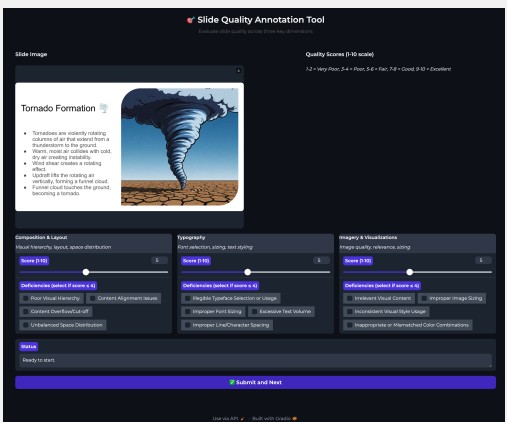

Figure 21: The UI of the user annotation interface built with Gradio.

**Scoring Dimensions and Rules**

**Three Scoring Dimensions (1-10 Scale)**

1. **Composition & Layout** - Evaluate overall layout, visual hierarchy, space distribution, element alignment
2. **Typography** - Evaluate font selection, size, readability, typesetting
3. **Imagery & Visualizations** - Evaluate image quality, content relevance, sizing, visual style consistency

**Scoring Scale**

- **9-10 (Excellent):** Outstanding quality, exquisite design
- **7-8 (Good):** Good quality with minor issues
- **5-6 (Fair):** Basically qualified but with some deficiencies
- **3-4 (Poor):** Obvious problems affecting overall effectiveness
- **1-2 (Very Poor):** Severely impacts slide quality

**Key Scoring Rules**

**Deficiency Annotation Rule**

> **Annotation Checklist**
>
> **IMPORTANT:** Only select specific deficiency types when a dimension scores $\leq 4$. If the score is $\geq 5$, no deficiency selection is required.

**Scoring Principles Checklist**

1. **Objective & Fair:** Score based on actual slide quality, not personal preferences
2. **Consistent Standards:** Maintain consistent scoring - similar quality slides get similar scores
3. **Comprehensive Assessment:** Consider all aspects of each dimension
4. **Relative Comparison:** Distribute scores reasonably within 1-10 range

**Operational Workflow**

1. Carefully examine slide content, design, and overall effectiveness

2. Score each of the three dimensions on a 1-10 scale
3. Select specific deficiency types only when score $\leq 4$
4. Submit annotation and proceed to next slide

---

**Preference Checklist**

**Content Quality**

- Is the information clear, accurate, and relevant?
- Are key points well-organized and logical?
- Is the text amount appropriate (avoiding information overload)?

**Visual Design**

- Is the layout balanced and aesthetically pleasing?
- Are colors harmonious and professional?
- Are fonts readable with appropriate sizing?
- Are images/charts high quality and clear?

**Readability**

- Is there sufficient contrast between text and background?
- Is content legible from a distance?
- Are important elements properly highlighted?

**Professionalism**

- Is the overall style consistent and professional?
- Are there any spelling or grammar errors?
- Are brand elements used appropriately?

**Effectiveness**

- Does it effectively convey the core message?
- Does it engage and maintain audience attention?
- Do visualizations aid comprehension?

## H  PROMPTS DESIGN

### H.1  TRAINING PROMPTS

**System Prompt**

A conversation between User and Assistant. The user asks a question, and the Assistant solves it. The assistant first thinks about the reasoning process in the mind and then provides the user with the answer. The reasoning process and answer are enclosed within ⟨think⟩ and ⟨/think⟩, and ⟨answer⟩ and ⟨/answer⟩ tags, respectively, i.e., ⟨think⟩ reasoning process here ⟨/think⟩⟨answer⟩ answer here ⟨/answer⟩.

**Scoring Task**

What is your overall rating on the quality of this slide? The rating should be a float between 1 and 10, rounded to two decimal places.

**Adjustment Task**

What are the design deficiencies in the provided slide? How can you improve the slide? Please answer by addressing the following questions for each category and provide detailed improvements suggestions. If there are no major deficiencies, please respond with "No major deficiencies found.".

1. **Composition & Layout:** What are the strengths and weaknesses of the slide's composition and layout? Consider alignment, balance, white space, and visual hierarchy. How would you rearrange, align, or resize elements to improve the overall structure and clarity?

2. **Typography:** How effective is the typography? Evaluate the font choices, readability, and visual hierarchy (titles, body text). What specific changes to fonts, sizes, or styling would you recommend to improve legibility and structure?

3. **Imagery & Visualizations:** Are the images and visualizations (e.g., charts, graphs) clear, relevant, and effective, or are they distracting and cluttered? What improvements, replacements, or simplifications would you make to better support the slide's message?

**Scoring Task**

Compare the two enhanced slides and determine which is superior, or if they are comparable. Evaluate them based on the following criteria: 1) Fidelity and Consistency: How well does the enhanced slide maintain the content, layout, and style of the original? 2) Overall Perceptual Quality: How visually appealing and free of artifacts is the slide? 3) Clarity and Readability: Are the text and visuals sharp and easy to understand? 4) Effectiveness of Communication: How effectively does the slide convey its intended message? Your answer should be exactly one of: Slide A, Slide B.

## H.2 GENERATION QUALITY EVALUATION PROMPTS

**Content Fidelity**

You are a meticulous presentation judge. Evaluate ONLY *Fidelity*—whether the slides faithfully reflect the provided reference without distortion or unsupported claims.

Inputs you may receive:
- slides[]: ordered slide images (index from 1)
- transcript: optional narration/notes text
- reference: gold context such as abstract, method/results bullets, or paper text
Rules:
1) Ground truth = reference (if provided). If no reference, judge fidelity via internal consistency across slides/transcript and obvious domain violations; do NOT use outside knowledge.
2) Reward: accurate statements aligned with reference; correct numbers, units, equations, method names; precise citations/attribution; clear reporting of limitations.
3) Penalize: factual errors; missing or altered key results; mislabeled axes/tables; cherry-picking; overstated conclusions; ambiguous provenance of images or data; uncredited reuse.
4) Be specific: cite where you found issues by slide number and short quote/description.
5) Ignore non-fidelity aspects (style, color, layout) unless they directly obscure truthfulness.

Score :
- 5: Fully faithful; no meaningful discrepancies.
- 4: Strong; minor omissions/nuances off, no material errors.
- 3: Mixed; some unclear or weakly supported claims; possible minor numeric/text mismatches.
- 2: Weak; notable inaccuracies, missing key qualifiers, or misleading visuals.
- 1: Unacceptable; major misrepresentation of method/results or pervasive errors.

## Content Clarity

Evaluate ONLY *Clarity*—how easily a typical informed audience can understand the content.

Inputs: slides[], transcript
Checkpoints:
- Language: concise phrasing, concrete nouns/verbs, defined terms/acronyms on first use.
- Structure: clear headings, bullets with parallel structure, limited per-slide ideas.
- Legibility: text size vs slide area, line length, contrast sufficient to read.
- Data clarity: axis labels, units, legends, footnotes; no orphaned numbers.
- Noise: redundant wording, clutter, irrelevant decorative elements that impede reading.
Score:
- 5: Effortless to follow; plain, precise language; every chart/table self-explanatory.
- 4: Mostly clear; minor verbosity or occasional undefined jargon.
- 3: Understandable with effort; several dense or ambiguous sections.
- 2: Hard to parse; frequent jargon, tiny text, missing labels.
- 1: Opaque; critical content unreadable or consistently unclear.

## Content Narrative

Evaluate ONLY *Narrative*—coherent story arc across the deck.

Inputs: slides[], transcript
Assess:
- Arc: clear beginning (motivation/problem), middle (method/approach), end (results, takeaway).
- Cohesion: transitions and signposting ("Problem $\rightarrow$ Approach $\rightarrow$ Results $\rightarrow$ Implications").
- Progression: each slide advances the story; no tangents; consistent voice tense/person.
- Framing: stakes, context, and audience relevance are stated and resolved.
Score:
- 5: Strong arc with seamless transitions and explicit takeaways.
- 4: Clear storyline; minor weak transitions or missing signposts.
- 3: Mixed; sections exist but connections are loose; some slides feel isolated.
- 2: Fragmented; jumps in logic; important steps missing.
- 1: Disjoint; no discernible arc or resolution.

## Content Engagement

Evaluate ONLY *Engagement*—how well the deck captures and sustains attention.

Inputs: slides[], transcript
Consider:
- Hooks: compelling opener (question, surprising stat, vivid example).
- Curiosity: rhetorical questions, progressive disclosure, contrasts before/after.
- Audience address: "you" framing, relevance cues, concrete examples.
- Interactivity prompts: checkpoints, polls, brief tasks (if present).
- Pacing signals: slide density balance; visual variety supporting attention.
Score:
- 5: Highly engaging; strong hook; sustained curiosity; clear calls to think or respond.
- 4: Generally engaging with a few lulls.
- 3: Moderately engaging; some interest, limited activation.
- 2: Low engagement; mostly static info-dump.
- 1: Not engaging; no hook or relevance cues.

## Design Elements

Evaluate ONLY *Elements*—choice and quality of visual assets (figures, icons, photos, charts, tables). Inputs: slides[]

Assess:
- Relevance: each element supports a stated point; no decorative-only clutter.
- Technical quality: resolution, cropping, anti-aliasing; no pixelation or watermarks.
- Semantics: correct chart type for data; legends/labels present; icons not misleading.
- Attribution: source captions when depicting external data/images.
- Consistency: style of icons/illustrations harmonious across slides.
 Score:
- 5: Elements are relevant, crisp, well-labeled, and consistently styled.
- 4: Minor issues but overall strong.
- 3: Mixed relevance or occasional low quality/label gaps.
- 2: Frequent low-res/mislabeled/misfit visuals.
- 1: Elements hinder understanding or are largely irrelevant.

## Design Layout

Evaluate ONLY *Layout*—placement, alignment, spacing, and balance.

 Inputs: slides[]
Assess:
- Grid  alignment: consistent margins; items align to an underlying grid.
- Spacing: sufficient whitespace; consistent gutters.
- Balance: visual weight distribution; avoids crowding.
- Flow: natural reading order (top→bottom, left→right).
 Score:
- 5: Clean grid, consistent alignment, ample whitespace, clear reading path.
- 4: Mostly strong; minor misalignments or occasional crowding.
- 3: Adequate but uneven spacing/alignment in places.
- 2: Frequent clutter, collisions, or confusing placement.
- 1: Chaotic layout; reading order unclear.

## Design Hierarchy

Evaluate ONLY *Hierarchy*—how typographic/visual cues convey importance and reading order. Inputs: slides[]
Assess:
- Typographic system: consistent levels (Title ¿ Subtitle ¿ Body ¿ Annotation).
- Emphasis: scale/weight/color/position guide attention.
- Grouping: headings correctly group related content.
- Consistency: same level looks the same across slides.
 Score:
- 5: Hierarchy is unmistakable; attention flows as intended.
- 4: Clear overall; occasional inconsistency.
- 3: Some hierarchy present but muddled.
- 2: Weak; competing focal points.
- 1: No discernible hierarchy.

## Design Color

Evaluate ONLY *Color*—palette coherence, accessibility contrast, and meaningful use. Inputs: slides[]
Assess:
- Palette: limited, harmonious palette; consistent use across slides.
- Contrast: text/background contrast sufficient.
- Semantics: colors map to meaning consistently.
- Legibility: avoid color-only encoding for critical distinctions.
 Score:

- 5: Cohesive palette, excellent contrast, consistent semantics.
- 4: Minor contrast or consistency issues.
- 3: Mixed; some low-contrast text or shifting category colors.
- 2: Frequent low contrast or confusing color use.

