# OpenReview forum: "Presenting a Paper is an Art: Self-Improvement Aesthetic Agents for Academic Presentations"
_ICLR.cc/2026/Conference — ICLR 2026 Poster_

### Official Review · Reviewer_A3kw · 2025-10-27

**Soundness:** 3
**Presentation:** 3
**Contribution:** 3
**Rating:** 8
**Confidence:** 4

**Summary:**

This paper introduces EvoPresent, a multi-agent framework designed to automate academic presentations by integrating a self-improvement loop. This loop is driven by PresAesth, a new multi-task RL model trained to evaluate presentation aesthetics and provide feedback. The authors validate their system using the new EvoPresent Benchmark, demonstrating that EvoPresent outperforms current methods and produces presentations with improved narrative coherence and visual design, approaching human-level quality.

**Strengths:**

EvoPresent goes beyond just slides; it supports multiple formats (videos, scripts, slides). The system aims to automate the entire act of presentation, not just the materials.

PresAesth is not a simple binary classifier. Its strength lies in its multi-task RL foundation, enabling it to provide (1) reliable aesthetic scoring, (2) specific defect identification for adjustment, and (3) comparative feedback. This rich, actionable feedback is far more useful for an agent than a simple score.

The paper introduces a much-needed, systematic benchmark EvoPresent Benchmark, which includes multimodal resources like slides, videos, and scripts.

The authors conduct extensive experiments with detailed analysis, which shows a huge workload with high quality.

**Weaknesses:**

The framework's evaluation is limited to static slide aesthetics. The Checker Agent and PresAesth model score aspects like layout and visualization, but they do not appear to evaluate the dynamic animation of slide elements. This is a significant omission, as the sequence, timing, and quality of animations are crucial for a coherent narrative and professional delivery. Consequently, the self-improvement loop cannot identify or correct errors in animation logic, such as elements appearing in the wrong order, which limits the final quality of the presentation playback.

The paper states in Appendix E that the data was labeled by "approximately 30 volunteers with design backgrounds" and "2-3 annotators". The paper does not report any inter-annotator agreement metrics to validate the consensus among annotators.

The self-improvement loop is focused exclusively on slide creation, while the video generation is treated as a separate, non-iterative task. The paper would be significantly stronger if the iterative optimization process also included feedback on the delivery itself, such as the virtual character's pacing, intonation, or synchronization with the dynamic slide content.

**Questions:**

For suggestions for authors, please refer to the Weakness part.

---

> ### Author Response · Authors · 2025-11-20
> **Rebuttal Part 1**
>
> **Dear Reviewer A3kw,**
>
> We appreciate the reviewer's thoughtful comments and recognition of ****EvoPresent’s ability to automate the entire act of presentation across multiple formats, the multi-task RL foundation of PresAesth that provides rich, actionable feedback, our systematic EvoPresent Benchmark, and the extensive experiments and high quality of work presented in the paper**.** We sincerely hope our response below can address your concerns.
>
> > Weakness 1: Lack of evaluation for dynamic animations
>
> We thank the reviewer for the insightful comments. We agree that evaluating dynamic presentation quality is essential for assessing the utility of an automated presentation system. In our work, we evaluate generated presentation videos using the Presentation Experience Evaluation in Figure 4(a-b) of the manuscript, which covers four dynamic dimensions: narrative coherence, temporal fluency, information pacing, and comprehension difficulty. EvoPresent achieves the best performance across all video-level metrics, indicating superior narrative flow and visual smoothness compared with existing systems.
>
> Regarding the concern about incorrect animation ordering, such issues typically stem from inconsistencies in narrative sequencing. In our pipeline, this is addressed at the source by the Storyline Agent (Section 3.1), which filters irrelevant content, infers logical dependencies between textual and visual elements, and constructs a coherent narrative order. This early-stage processing prevents misaligned element ordering during rendering. To further validate this capability, we select 50 samples with intentionally shuffled narrative order and evaluate narrative quality before and after reconstruction using overall perplexity (PPL) and the four dynamic metrics. As shown in Table 8, the Storyline Agent effectively restores coherent sequencing and improves overall narrative consistency. As part of future work, we plan to incorporate timeline consistency and additional dynamic quality checks into the PresAesth framework, enabling more detailed temporal evaluation and further improving the robustness of automated presentation generation.
>
> Table 8: Evaluation on 50 shuffled-order samples.
> | Metric | Before Storyline Agent | After Storyline Agent |
> | :--- | :---: | :---: |
> | Narrative Coherence ↑ | 0.62 | 0.85 |
> | Temporal Fluency ↑ | 0.68 | 0.89 |
> | Information Pacing ↑ | 0.60 | 0.83 |
> | Comprehension ↑ | 0.72 | 0.93 |
> | PPL ↓ | 38.5 | 23.4 |
>
> > Weakness 2: Insufficient reporting on annotation quality
>
> We appreciate this thoughtful suggestion and fully agree that annotator consistency and transparency are essential for building a reliable aesthetic benchmark. As shown in Appendix G.1, we provide the visual interfaces of our annotation platform and task-specific checklists, and Appendix E.1 presents representative annotated examples to clarify the workflow and decision criteria. To further assess annotator consistency across different tasks, we use Inter-Class Correlation (ICC) [3] for continuous rating tasks and Fleiss’ Kappa [4] for discrete classification tasks. ICC measures consistency in numerical ratings for aesthetic and presentation-quality scoring, while Fleiss’ Kappa evaluates agreement in categorical tasks such as defect detection and pairwise comparison. As shown in Table 9, annotators achieve high agreement across all tasks, with low variance and stable annotation behavior. These results indicate that our benchmark delivers reliable subjective annotations and provides a solid foundation for training and evaluating PresAesth.
>
> Table 9: Inter-annotator consistency across different annotation tasks.
> | Task | Metric | Consistency Score |
> | :--- | :---: | :---: |
> | Aesthetic Score (1--10) | ICC | 0.92 |
> | Presentation Quality Score (0-5) | ICC | 0.95 |
> | Defect Detection | Fleiss' Kappa | 0.80 |
> | Pairwise Aesthetic Comparison | Fleiss' Kappa | 0.83 |
>
> [3] Shrout, P. E., & Fleiss, J. L. (1979). Intraclass correlations: uses in assessing rater reliability. Psychological bulletin, 86(2), 420.
>
> [4] Fleiss, J. L. (1971). Measuring nominal scale agreement among many raters. Psychological bulletin, 76(5), 378.

---

> > ### Author Response · Authors · 2025-11-20
> > **Rebuttal Part 2**
> >
> > > Weakness 3: Self-improvement loop ignores delivery / video aspects
> >
> > We appreciate the reviewer’s insightful suggestion. We agree that extending the self-improvement loop to the video level could further enhance the system’s performance in real presentation settings. In the current version, video generation remains a non-iterative component primarily due to computational considerations. Iterative optimization of presentation videos introduces substantial overhead, as Table 10 shows that video-level iteration requires roughly five times the inference time of slide-based refinement, making it challenging to meet the efficiency requirements of typical use scenarios. Furthermore, as demonstrated in the Presentation Experience Evaluation in Section 5.1, the system already achieves strong performance in narrative coherence, pacing, temporal smoothness, and comprehensibility without video-level iteration, suggesting that static content refinement is sufficient to support high-quality video presentations. Looking ahead, we plan to explore integrating video presentation into the reinforcement-driven self-improvement loop, for example by incorporating prosody evaluators and timeline alignment checks. We include a discussion of these potential extensions in the revised manuscript to clarify the current design choices and outline future pathways for video-level iterative optimization.
> >
> > Table 10: Inference cost comparison between slide and video-based iterative refinement
> > | Method | Avg. Inference Time (min) | Relative Cost |
> > | :--- | :---: | :---: |
> > | Slide-based Iterative Refinement | 5.56 | $1.0\times$ |
> > | Video-based Iterative Refinement | 26.0 | $4.7\times$ |
> >
> > **Best regards,**
> >
> > **Authors**

---

### Official Review · Reviewer_JPj1 · 2025-10-28

**Soundness:** 3
**Presentation:** 3
**Contribution:** 2
**Rating:** 2
**Confidence:** 4

**Summary:**

This paper introduces EvoPresent, a novel self-improving multi-agent framework designed to automate the creation of high-quality academic presentations. The system employs specialized agents for storyline construction, content enrichment, and visual design, all coordinated within a draft-feedback-refinement loop. A key innovation is PresAesth, a multi-task reinforcement learning model that provides aesthetic scoring, defect adjustment, and comparative feedback, enabling iterative self-improvement even with limited human preference data. The authors also contribute the EvoPresent Benchmark for systematic evaluation. Experiments demonstrate that the framework surpasses existing methods in generating presentations with superior narrative coherence and visual appeal, highlighting the critical role of high-quality aesthetic feedback for effective self-correction.

**Strengths:**

1. The paper introduces a novel and interesting task, which indeed leaves ample room for further exploration and development.

2. The overall aesthetic quality of the paper is quite good — the figures, charts, and tables are all well-designed and visually appealing.

3. The supplementary materials are sufficient and help present the work in a more comprehensive and detailed manner.

**Weaknesses:**

1. First of all, the dataset size seems quite limited. For Presentation Generation Quality, there are only about 650 samples, and for Aesthetic Awareness, just around 2,000 slides. That’s quite small for training or evaluating models of this scale.

2. Regarding the checker agent, it’s not clear how the threshold for evaluation is determined. Moreover, since different slides may have very different design intents — some emphasizing clarity, others emphasizing key visual highlights — are these aesthetic thresholds the same across all slides, or adaptive to content type?

3. When it comes to the collaboration among agents, it seems the paper doesn’t discuss what happens when one agent makes an error. How do you detect and handle such errors in real time, or mitigate their impact to improve overall cooperation efficiency?

4. Minor issue: in line 209, it seems that item (ii) is missing.

5. I’m a bit confused about how Generation Quality and Aesthetic Awareness are differentiated. You mentioned that Generation Quality is evaluated using your own trained PresAesth model to obtain aesthetic scores — but how exactly are these scores different from those used in Aesthetic Awareness?

6. During the data annotation process, although you mentioned using multiple annotators, it seems you didn’t assess their inter-annotator agreement (e.g., Fleiss’ Kappa). Given that this task heavily depends on subjective annotation quality, more details about annotator backgrounds, collaboration protocols, and agreement levels would significantly improve credibility. Also, showing a few annotation examples would help demonstrate data reliability.

7. In Figure 13, some human faces are shown without blurring. This potentially raises ethical concerns regarding privacy and data usage.

8. Overall, the task itself is interesting and has an innovative motivation. However, the technical side feels more like assembling multiple existing agents and applying reinforcement learning as a wrapper. There’s little algorithmic novelty, and the data annotation process doesn’t clearly offer a distinctive contribution. The demos look nice, but the practical value remains somewhat unclear, and the work seems limited mainly to the computer science domain.

**Questions:**

See above

---

> ### Author Response · Authors · 2025-11-20
> **Rebuttal Part 1**
>
> **Dear Reviewer JPj1,**
>
> We appreciate the reviewer's thoughtful comments and recognition of **our novel and interesting task, the aesthetic quality of our figures, and the comprehensiveness of our supplementary materials**. We sincerely hope our response below can address your concerns.
>
> > Weakness 1: First of all, the dataset size seems quite limited. For Presentation Generation Quality, there are only about 650 samples, and for Aesthetic Awareness, just around 2,000 slides. That’s quite small for training or evaluating models of this scale.
>
> We thank the reviewer for their attention to the EvoPresent Benchmark and would like to further clarify its composition and coverage. For the presentation generation quality evaluation task, the 650 samples represent 650 complete presentation files, rather than individual slides. Each presentation comprises around 20 slides, which brings the overall evaluation coverage to **13,000** real academic slides. As shown in Fig. 3(a–c) of the manuscript, their topic and domain distribution ensures strong diversity and representativeness. For the Aesthetic Awareness task, we construct 2,000 slide triplets, each consisting of poor, base, and good versions of the same slide, resulting in **6,000** annotated aesthetic samples. The detailed structure and construction process of this dataset is described in Section 4.2. Moreover, the results (Table 3 and Appendix D.2 of the manuscript) show that our RL training consistently improves the performance, showing the **effectiveness of our training data.** In addition, the fact that our multi-task RL training achieves strong robustness and generalization even with a moderate amount of aesthetic annotations further supports **the effectiveness of our training paradigm and highlights its sample efficiency.**
>
> > Weakness 2: Regarding the checker agent, it’s not clear how the threshold for evaluation is determined. Moreover, since different slides may have very different design intents — some emphasizing clarity, others emphasizing key visual highlights — are these aesthetic thresholds the same across all slides, or adaptive to content type?
>
> We thank the reviewer for the valuable comments and provide additional clarification regarding the rationale behind the aesthetic threshold. As shown in Figure 7 of the manuscript, the system adopts an aesthetic threshold of 8.0. We further conduct validation experiments across several candidate thresholds, and the results appear in Table 3. Lower thresholds reduce the number of refinement iterations and improve efficiency, but they also cause a noticeable drop in final aesthetic quality. Higher thresholds substantially increase iteration counts and inference cost, while offering only marginal aesthetic gains. Considering both aesthetic improvement and computational overhead, a threshold of 8.0 provides a favorable balance between quality and efficiency.
>
> Regarding the applicability of this threshold, we would like to clarify that PresAesth is designed to provide a **unified assessment of overall slide aesthetics**, rather than relying on scoring rules that depend on specific design intents. The benchmark dataset includes a broad range of common academic slide types, which enables the model to learn a consistent and robust aesthetic judgment standard. To further examine this, we group slides in the EvoPresent Benchmark into four content-structure categories (text-only, image-only, text-dominant mixed, and image-dominant mixed), and compute the correlation and error between model predictions and human annotations for each category. As shown in Table 4, the results are highly comparable across categories, suggesting that the model maintains stable and consistent scoring behavior under different design forms. Therefore, using a single threshold during inference provides reliable quality control, without the need to set different thresholds for different design types. We have added further explanations and supporting results in the revised manuscript.
>
> Table 3: Comparison of different aesthetic thresholds.
> | Threshold ($S_{th}$) | Final Aesthetic Score $\uparrow$ | Avg. Iterations $\downarrow$ |
> | :--- | :---: | :---: |
> | 6.5 | 7.36 | 4.8 |
> | 7.0 | 7.42 | 5.0 |
> | 7.5 | 7.80 | 5.8 |
> | **8.0** | **8.05** | **6.2** |
> | 8.5 | 8.09 | 8.9 |
> | 9.0 | 8.10 | 10.5 |
>
> Table 4: Aesthetic prediction consistency across different slide designs.
> | Slide Type | Pearson $r$ $\uparrow$ | MAE $\downarrow$ |
> | :--- | :---: | :---: |
> | Text-only slides | 0.858 | 1.325 |
> | Image-only slides | 0.854 | 1.330 |
> | Text-dominant mixed slides | 0.847 | 1.316 |
> | Image-dominant visual slides | 0.853 | 1.310 |

---

> > ### Author Response · Authors · 2025-11-20
> > **Rebuttal Part 2**
> >
> > > Weakness 3: When it comes to the collaboration among agents, it seems the paper doesn’t discuss what happens when one agent makes an error. How do you detect and handle such errors in real time, or mitigate their impact to improve overall cooperation efficiency?
> >
> > We thank the reviewer for the valuable feedback. We agree that timely detection and handling of potential errors in a multi-agent system is essential for maintaining workflow stability, and EvoPresent is designed around this requirement. During generation, the textual content, narrative scripts, and layout of each slide are assessed by the PresAesth module within the Checker Agent. Beyond aesthetic scoring, PresAesth identifies structural and semantic issues such as layout imbalance, narrative inconsistencies, and missing images (see Appendix H.1 for details). As described in Section 3.1, the Storyline Agent also performs filtering, deduplication, and consistency checks during content extraction, reducing upstream noise and alleviating the burden on downstream agents. With the system’s iterative closed-loop design, detected errors are immediately returned to the responsible agent for targeted correction, preventing error propagation across the pipeline (see Figure 7 and Table 3 of the manuscript). To further evaluate the system’s error detection capability, we sample 450 defective slides from the aesthetic awareness benchmark, covering content, layout, and visual error types. As shown in Table 5, PresAesth achieves high detection accuracy across all categories and substantially outperforms the general-purpose model GPT-4o, demonstrating its reliability in identifying structural, semantic, and visual defects.
> >
> > Table 5: Detection accuracy across different error types for GPT-4o and PresAesth.
> > | Error Type | GPT-4o Detection Rate (%) | PresAesth Detection Rate (%) |
> > | :--- | :---: | :---: |
> > | Content Errors | 82.3% | 95.2% |
> > | Layout Errors | 85.6% | 96.1% |
> > | Visual Errors | 88.9% | 97.5% |
> > | Macro Avg. | 85.6% | 96.3% |
> >
> > > Weakness 4: Minor issue: in line 209, it seems that item (ii) is missing.
> >
> > We appreciate the reviewer for pointing out the missing item; it has been corrected in the revised manuscript.
> >
> > > Weakness 5: I’m a bit confused about how Generation Quality and Aesthetic Awareness are differentiated. You mentioned that Generation Quality is evaluated using your own trained PresAesth model to obtain aesthetic scores — but how exactly are these scores different from those used in Aesthetic Awareness?
> >
> > We appreciate this valuable comment and would like to clarify that Generation Quality and Aesthetic Awareness are **two distinct and independent evaluation dimensions with different objectives and assessment criteria**. As defined in Section 4.1, Generation Quality Task measures the overall quality of the final presentation produced by the full EvoPresent system after completing the multi-agent pipeline. It focuses on narrative coherence, visual layout design, and overall aesthetic appeal, and is evaluated using the 12 complementary content and design metrics listed in Table 2 of the manuscript, which collectively reflect system-level performance in real generation tasks. In contrast, Aesthetic Awareness Task evaluates the capability of our RL-based aesthetic model, PresAesth, independent of the final generated output. We construct a dedicated benchmark for this purpose (Table 3 of the manuscript), assessing PresAesth on three subtasks: aesthetic score prediction, defect detection, and pairwise comparison between slides of different aesthetic quality. These tasks use entirely different metrics, including MSE for score prediction and accuracy for defect detection and pairwise judgments.
> >
> > In summary, Generation Quality Task evaluates the end-to-end performance of the EvoPresent system, whereas Aesthetic Awareness Task measures the aesthetic assessment ability of PresAesth. They differ in goals, metrics, and datasets, and together provide a comprehensive view of system-level generation quality and aesthetic reasoning ability. We highlight this distinction in the revised manuscript.

---

> > > ### Author Response · Authors · 2025-11-20
> > > **Rebuttal Part 3**
> > >
> > > > Weakness 6: During the data annotation process, although you mentioned using multiple annotators, it seems you didn’t assess their inter-annotator agreement (e.g., Fleiss’ Kappa). Given that this task heavily depends on subjective annotation quality, more details about annotator backgrounds, collaboration protocols, and agreement levels would significantly improve credibility. Also, showing a few annotation examples would help demonstrate data reliability.
> > >
> > > We appreciate this thoughtful suggestion and fully agree that annotator consistency and transparency are essential for building a reliable aesthetic benchmark. As shown in Appendix G.1, we provide the visual interfaces of our annotation platform and task-specific checklists, and Appendix E.1 presents representative annotated examples to clarify the workflow and decision criteria. To further assess annotator consistency across different tasks, we use Inter-Class Correlation [3] (ICC) for continuous rating tasks and Fleiss’ Kappa [4] for discrete classification tasks. ICC measures consistency in numerical ratings for aesthetic and presentation-quality scoring, while Fleiss’ Kappa evaluates agreement in categorical tasks such as defect detection and pairwise comparison. As shown in Table 6, annotators achieve high agreement across all tasks, with low variance and stable annotation behavior. These results indicate that our benchmark delivers reliable subjective annotations and provides a solid foundation for training and evaluating PresAesth.
> > >
> > > [3] Shrout, P. E., \& Fleiss, J. L. (1979). Intraclass correlations: uses in assessing rater reliability. Psychological bulletin, 86(2), 420.
> > >
> > > [4] Fleiss, J. L. (1971). Measuring nominal scale agreement among many raters. Psychological bulletin, 76(5), 378.
> > >
> > > Table 6: Inter-annotator consistency across different annotation tasks.
> > > | Task | Metric | Consistency Score |
> > > | :--- | :---: | :---: |
> > > | Aesthetic Score (1--10) | ICC | 0.92 |
> > > | Presentation Quality Score (0-5) | ICC | 0.95 |
> > > | Defect Detection | Fleiss' Kappa | 0.80 |
> > > | Pairwise Aesthetic Comparison | Fleiss' Kappa | 0.83 |
> > >
> > > > Weakness 7: In Figure 13, some human faces are shown without blurring. This potentially raises ethical concerns regarding privacy and data usage.
> > >
> > > We also thank the reviewer for highlighting the privacy issue, and the relevant region in Figure 13 has been appropriately blurred.

---

> > > > ### Author Response · Authors · 2025-11-20
> > > > **Rebuttal Part 4**
> > > >
> > > > > Weakness 8: Overall, the task itself is interesting and has an innovative motivation. However, the technical side feels more like assembling multiple existing agents and applying reinforcement learning as a wrapper. There’s little algorithmic novelty, and the data annotation process doesn’t clearly offer a distinctive contribution. The demos look nice, but the practical value remains somewhat unclear, and the work seems limited mainly to the computer science domain.
> > > >
> > > > Thank you for the reviewer’s valuable comments. While EvoPresent benefits from a well-integrated architecture, its main contribution lies in the algorithmic designs that address the largely unexplored problem of aesthetic-aware academic presentation generation. The key components are summarized below: We appreciate the reviewer’s feedback. We would like to clarify that the contribution of EvoPresent goes beyond system integration. Our work introduces algorithmic designs specifically tailored to the largely unexplored problem of aesthetic-aware academic presentation generation. **The key contributions are summarized below:**
> > > >
> > > > 1. **We introduce the first unified multi-task aesthetic framework.** Prior aesthetic studies focus on isolated tasks (e.g., scoring), which cannot support the multi-dimensional design reasoning required in presentation generation. Our formulation jointly learns aesthetic scoring, defect localization, and version comparison, enabling shared aesthetic principles to emerge across tasks and yielding clear gains over single-task baselines (Table 5 of the manuscript). We further design a multi-dimensional aesthetic reward mechanism (Sec. 3.2 of the manuscript) that transforms subjective aesthetic judgments into optimizable signals. To our knowledge, such a reward structure has not been explored in existing aesthetic modeling or RL research.
> > > > 2. **We propose an iterative self-improvement agent paradigm.** Existing self-improvement techniques mainly focus on textual tasks, while iterative optimization for visual aesthetics remains challenging due to its subjectivity. As shown in Fig. 7(b)  of the manuscript, even strong general-purpose models such as GPT-4o struggle to self-correct on aesthetic tasks, indicating that advanced reasoning alone is insufficient for visual refinement. In contrast, our approach leverages precise and executable aesthetic feedback from PresAesth, enabling effective iterative improvement and supporting the development of aesthetic-aware agent workflows.
> > > > 3. **We construct the first large-scale benchmark for academic presentations aesthetics.** Previous research lacks any dataset that supports multi-task aesthetic reasoning for presentation slides. Our Evopresent benchmark fills this void by offering systematically annotated, multi-dimensional aesthetic labels, providing a reliable basis for model training and comprehensive evaluation. It also establishes a reusable resource for future research in this domain.
> > > >
> > > > In terms of practical utility, extensive experiments and user studies demonstrate that EvoPresent consistently outperforms existing methods in both content quality and visual design, as shown in Figures 4–6 and Tables 2–3 of the manuscript. The user study in Figure 6 further confirms clear preference for our system, and Appendix D.3 provides additional qualitative examples.
> > > >
> > > > In terms of application generalization, EvoPresent is not restricted to computer science papers. Since the pipeline and PresAesth’s training objectives do not depend on domain-specific semantics, the system naturally generalizes to other academic and professional documents. To further verify its generalization capability, we additionally evaluate the framework on 100 cross-domain documents (e.g., engineering reports and management documents) without fine-tuning. The system achieves performance comparable to that on research papers, showing stable content structuring and visual design. As shown in Table 7, PresAesth also maintains high accuracy on aesthetic scoring, defect detection, and pairwise comparison, indicating strong cross-domain generalization. All additional experimental results are included in the revised manuscript.
> > > >
> > > > Table 7: Cross-domain performance of PresAesth on 100 cross-domain documents.
> > > > | Task | Metric | Accuracy on cross-domain documents |
> > > > | :--- | :---: | :---: |
> > > > | Aesthetic Scoring | MSE ↓ | 1.35 |
> > > > | Defect Detection | Accuracy ↑ | 78.5% |
> > > > | Pairwise Comparison | Accuracy ↑ | 90.7% |
> > > >
> > > > **Best regards,**
> > > >
> > > > **Authors**

---

> ### Author Response · Authors · 2025-11-26
> **Gentle Reminder**
>
> **Dear Reviewer JPj1,**
>
> Thank you very much for your careful review and constructive comments. Your comments are highly valuable to the improvement of our work. We have carefully addressed your comments regarding dataset scale, evaluation thresholds, multi-agent collaboration, and annotation consistency, providing detailed analyses and additional experiments in our rebuttal.
>
> With less than one week remaining in the discussion period, we sincerely invite you to review our responses. We hope that these clarifications effectively resolve your concerns, and we would greatly welcome any further suggestions you may have.
>
> Thank you again for your time and professional insight.
>
> Sincerely,
>
> **The Authors**

---

### Official Review · Reviewer_SjBg · 2025-10-30

**Soundness:** 3
**Presentation:** 3
**Contribution:** 3
**Rating:** 6
**Confidence:** 4

**Summary:**

This paper introduces EvoPresent, a multi-agent framework for automatically generating academic presentations from research papers. The system consists of four sequential agents: (1) Storyline Agent for content extraction and narrative construction, (2) Scholar Agent for content enrichment, (3) Design Agent for layout and rendering in HTML format, and (4) Checker Agent for iterative quality improvement. Central to the approach is PresAesth, a multi-task reinforcement learning model based on Qwen-2.5-VL-7B, trained via Group Relative Policy Optimization (GRPO) to perform aesthetic scoring (1-10 scale), defect adjustment, and pairwise comparison. The authors construct the EvoPresent Benchmark with 650 annotated presentations from top AI conferences and 2,000 slide pairs for aesthetic evaluation. Experiments demonstrate that EvoPresent outperforms existing methods (PPTAgent, PresentAgent, Paper2Poster) and end-to-end LLM approaches (GPT-4o, GPT-5, Claude-4-Sonnet) across multiple metrics, achieving aesthetic scores of 8.05/10 (approaching the 8.50 oracle score) and 87.8% accuracy on aesthetic comparisons (versus 77.1% for GPT-4o). The iterative self-improvement mechanism enables the system to reach target quality in fewer iterations compared to baselines.

**Strengths:**

1、The four-agent architecture is well-structured with clear separation of concerns. The iterative draft-feedback-refinement loop with reversion to best previous version demonstrates thoughtful engineering to prevent quality degradation.

2、Automating academic presentation generation is a relevant task with clear applications in research dissemination.

3、Comprehensive evaluation including quantitative metrics (perplexity, ROUGE-L, layout balance), fine-grained content/design scores, human preference studies, and ablations. The results consistently demonstrate improvements over baselines.

**Weaknesses:**

1、 While the system integration is solid, the core components rely on standard techniques—sequential agent architectures, GRPO for preference learning, and VLM-based evaluation. The main contribution is engineering integration rather than algorithmic innovation. The multi-task RL formulation is relatively straightforward.

2、The reliance on HTML rendering for "maximal control and flexibility" limits practical utility, as most researchers work with PowerPoint or PDF formats. While conversion is mentioned, the fidelity and editability of converted presentations are not evaluated.

**Questions:**

see weakness

---

> ### Author Response · Authors · 2025-11-20
> **Rebuttal Part 1**
>
> **Dear Reviewer SjBg,**
>
> We appreciate the reviewer's thoughtful comments and recognition of the **clear division of roles in our agent architecture, the effectiveness of our self-improvement mechanism, and the comprehensive evaluation demonstrating improvements over baselines.**  We sincerely hope our response below can address your concerns.
>
> > Weakness 1: While the system integration is solid, the core components rely on standard techniques—sequential agent architectures, GRPO for preference learning, and VLM-based evaluation. The main contribution is engineering integration rather than algorithmic innovation. The multi-task RL formulation is relatively straightforward.
>
> We appreciate the reviewer’s feedback. We would like to clarify that the contribution of EvoPresent goes beyond system integration. Our work introduces algorithmic designs specifically tailored to the largely unexplored problem of aesthetic-aware academic presentation generation. The key components are summarized below:
>
> 1. **We introduce the first unified multi-task aesthetic framework.** Prior aesthetic studies focus on isolated tasks (e.g., scoring), which cannot support the multi-dimensional design reasoning required in presentation generation. Our formulation jointly learns aesthetic scoring, defect localization, and version comparison, enabling shared aesthetic principles to emerge across tasks and yielding clear gains over single-task baselines (Table 5 of the manuscript). We further design a multi-dimensional aesthetic reward mechanism (Sec. 3.2 of the manuscript) that transforms subjective aesthetic judgments into optimizable signals. To our knowledge, such a reward structure has not been explored in existing aesthetic modeling or RL research.
> 2. **We propose an iterative self-improvement agent paradigm.** Existing self-improvement techniques mainly focus on textual tasks, while iterative optimization for visual aesthetics remains challenging due to its subjectivity. As shown in Fig. 7(b) of the manuscript, even strong general-purpose models such as GPT-4o struggle to self-correct on aesthetic tasks, indicating that advanced reasoning alone is insufficient for visual refinement. In contrast, our approach leverages precise and executable aesthetic feedback from PresAesth, enabling effective iterative improvement and supporting the development of aesthetic-aware agent workflows.
> 3. **We construct the first large-scale benchmark for academic presentations aesthetics.** Previous research lacks any dataset that supports multi-task aesthetic reasoning for presentation slides. Our Evopresent benchmark fills this void by offering systematically annotated, multi-dimensional aesthetic labels, providing a reliable basis for model training and comprehensive evaluation. It also establishes a reusable resource for future research in this domain.

---

> > ### Author Response · Authors · 2025-11-20
> > **Rebuttal Part 2**
> >
> > > Weakness 2: The reliance on HTML rendering for "maximal control and flexibility" limits practical utility, as most researchers work with PowerPoint or PDF formats. While conversion is mentioned, the fidelity and editability of converted presentations are not evaluated.
> >
> > We sincerely appreciate the reviewer’s comments and would like to further clarify the motivation and practicality of using HTML rendering. We understand that PowerPoint and PDF formats are more common in real research workflows. As noted in Appendix D.4 and Section 3.1, our system already supports generation and conversion across multiple formats, including HTML, Markdown, and PPTX; HTML files can also be losslessly exported to PDF via standard browser functionality, ensuring compatibility with mainstream academic workflows. To verify practicality, we additionally evaluate the outputs along two dimensions: consistency and editability.
> >
> > 1. **Consistency Evaluation:** We sample 600 HTML presentations from the EvoPresent Benchmark and compare them with their corresponding PPT and PDF versions using two widely adopted metrics, SSIM (structural similarity) [1] and LPIPS (perceptual similarity)[2]. As shown in Table 1, the converted PPT/PDF files achieve average SSIM scores above 0.96 and LPIPS scores below 0.10, indicating that layout structure, typography, and spatial composition are highly preserved during conversion.
> > 2. **Editability Evaluation:** We further examine the structural editability of the converted PPT files, including whether text remains in editable text boxes, images remain as independent objects, layout hierarchies are preserved, and style attributes (e.g., fonts, colors, sizes) remain adjustable. As shown in Table 2, about 98% of text boxes remain editable, all images are preserved as independent objects, and most layout and style structures are retained. These results demonstrate that the converted slides not only preserve the visual fidelity of the original outputs but also support the content and layout refinements required in practical research workflows.
> >
> > Table 1: Evaluation of structural and perceptual consistency across different conversion formats.
> > | Conversion Format | SSIM ↑ | LPIPS ↓ |
> > | :--- | :---: | :---: |
> > | HTML → PPT | 0.963 | 0.10 |
> > | HTML → PDF | 0.987 | 0.03 |
> > | HTML → Markdown | 0.974 | 0.08 |
> >
> > Table 2: Editability preservation of converted presentation.
> > | Evaluation Dimension | Retention Rate (%) |
> > | :--- | :---: |
> > | Text-box Editability | 98.5% |
> > | Image Object Preservation | 100% |
> > | Layout Hierarchy Integrity | 98.3% |
> > | Style Attribute Editability | 96.1% |
> >
> > [1] Wang, Z., Bovik, A. C., Sheikh, H. R., \& Simoncelli, E. P. (2004).
> > Image quality assessment: From error visibility to structural similarity.
> > IEEE Transactions on Image Processing, 13(4), 600–612.
> >
> > [2] Zhang, R., Isola, P., Efros, A. A., Shechtman, E., \& Wang, O. (2018).
> > The Unreasonable Effectiveness of Deep Features as a Perceptual Metric.
> > Proceedings of the IEEE Conference on Computer Vision and Pattern Recognition (CVPR), 586–595.
> >
> > **Best regards,**
> >
> > **Authors**

---

> ### Author Response · Authors · 2025-11-26
> **Gentle Reminder**
>
> **Dear Reviewer SjBg,**
>
> Thank you very much for your careful review and constructive comments. Your insights are highly valuable to the improvement of our work. We have carefully addressed your comments regarding the system’s component design and the HTML-based rendering workflow. These points have been thoroughly clarified and supplemented in our rebuttal, along with additional analyses and supporting evidence.
>
> With less than one week remaining in the discussion period, we sincerely invite you to review our responses. We hope that these clarifications effectively resolve your concerns, and we would greatly welcome any further suggestions you may have.
>
> Thank you again for your time and thoughtful consideration.
>
> Best regards,
>
> **The Authors**

---

### Author Response · Authors · 2025-11-21
**General response to all reviewers**

For clarity and simplicity, we will refer to Reviewers SjBg, JPj1, and A3kw as R1, R2, and R3, respectively, in the following response.

We sincerely thank all reviewers for their thoughtful and constructive feedback. We are encouraged by their recognition of the key contributions and strengths of our work. In particular, we thank the reviewers for highlighting the novelty of the academic presentation generation task and its practical value in automating multiple formats **(R1, R2, R3)**. We also appreciate their recognition of the clear role division in our multi-agent architecture, the actionable feedback provided by the PresAesth multi-task RL framework, and the effectiveness of the iterative self-improvement mechanism **(R1, R3)**. Moreover, we are grateful for their acknowledgement of the systematic construction of the EvoPresent Benchmark and the extensive, high-quality experimental evaluation **(R1, R3)**. Finally, we are pleased to see reviewers endorse the high aesthetic quality of our paper's figures and the comprehensiveness of our supplementary materials **(R1, R2, R3)**. In response to the reviews, we have updated the PDF and highlighted all modifications in blue in the revised manuscript.

We again thank all reviewers for their efforts, and we would be very happy to further clarify any remaining questions and continuously improve our paper.

---

### Author Response · Authors · 2025-12-03
**Summary of Rebuttals to Area Chair--Part 1**

Dear Area Chairs, Senior Area Chairs,

We sincerely appreciate your efforts in addressing the recent issue involving the unintended disclosure of reviewer identities. We fully acknowledge and support the measures taken to uphold the integrity of the review process and the fairness of the broader research ecosystem, and we are grateful for the diligence and responsibility demonstrated throughout this process.

To facilitate an efficient understanding of the key points in the rebuttal for submission 3794 (EvoPresent), we provide below a concise summary of the strengths recognized by the reviewers, along with the key contributions of our work and the main updates and points of agreement reached during the rebuttal exchange.

##  1. Summary of Strengths and Contributions
For clarity, we refer to reviewers **SjBg, JPj1, and A3kw** as **R1, R2, and R3**, respectively.

We thank all reviewers for their careful evaluation and constructive feedback. The reviewers consistently recognized the novelty and practical value of our work on automated academic presentation generation (**R1, R2, R3**). They acknowledged the clear role division and effective coordination within our four-agent architecture (**R1**), emphasized the importance of the actionable aesthetic feedback provided by PresAesth for iterative refinement (**R1, R3**), appreciated the overall presentation quality and completeness of the supplementary experiments (**R2**), and recognized the systematic value of the EvoPresent Benchmark for multimodal academic presentation evaluation (**R1, R3**).

Regarding contributions, EvoPresent addresses the under-explored problem of aesthetic-aware academic presentation generation through the following components:
1. **We introduce the first unified multi-task aesthetic framework.** Prior aesthetic studies focus on isolated tasks (e.g., scoring), which cannot support the multi-dimensional design reasoning required in presentation generation. Our formulation jointly learns aesthetic scoring, defect localization, and version comparison, enabling shared aesthetic principles to emerge across tasks and yielding clear gains over single-task baselines (Table 5 of the manuscript). We further design a multi-dimensional aesthetic reward mechanism (Sec. 3.2 of the manuscript) that transforms subjective aesthetic judgments into optimizable signals. To our knowledge, such a reward structure has not been explored in existing aesthetic modeling or RL research.

2. **We propose an iterative self-improvement agent paradigm.**  Existing self-improvement techniques mainly focus on textual tasks, while iterative optimization for visual aesthetics remains challenging due to its subjectivity. As shown in Fig. 7(b) of the manuscript, even strong general-purpose models such as GPT-4o struggle to self-correct on aesthetic tasks, indicating that advanced reasoning alone is insufficient for visual refinement. In contrast, our approach leverages precise and executable aesthetic feedback from PresAesth, enabling effective iterative improvement and supporting the development of aesthetic-aware agent workflows.

3. **We construct the first large-scale benchmark for academic presentations aesthetics.** Previous research lacks any dataset that supports multi-task aesthetic reasoning for presentation slides. Our Evopresent benchmark fills this void by offering systematically annotated, multi-dimensional aesthetic labels, providing a reliable basis for model training and comprehensive evaluation. It also establishes a reusable resource for future research in this domain.

## 2. Rebuttal and Discussion Summary
1. Reviewer **A3kw**  provides positive feedback on the overall system design, agent collaboration, and iterative self-improvement process, and offers constructive suggestions regarding the narrative order of dynamic animations and the iterative strategy for video generation. In our rebuttal, we conduct additional analysis and validation to address these points.  (1) For narrative order, we test 50 shuffled examples and find that the Storyline Agent improves narrative coherence by 37% and reduces perplexity by about 39%, showing that the system consistently enhances the restoration and organization of narrative structure. (2) For video iterative optimization, our comparison shows that video-level refinement is about 4.7 times more expensive than slide-level refinement. Under the current slide-level iteration setup, the system already produces temporally smooth and high-quality videos efficiently, highlighting the practical advantages of the static iteration scheme and providing a strong basis for future video-level extensions.   In addition,  Reviewer A3kw **responds positively during the rebuttal exchange, expresses strong agreement with our clarifications**, and confirms that all previously raised concerns are fully resolved.

---

### Author Response · Authors · 2025-12-03
**Summary of Rebuttals to Area Chair--Part 2**

2. Reviewer **SjBg** recognizes the clarity of the architecture, the well-defined division of responsibilities among the four agents, and the effectiveness of the iterative self-improvement mechanism, and raises constructive suggestions regarding format-conversion fidelity and agent module design. In the rebuttal, we provide additional empirical support to address these points:

   **(1) For format conversion**, we conduct a systematic evaluation of structural consistency and editability from HTML to PDF/PPTX. The results show that SSIM （structural similarity）scores remain above 0.95, LPIPS (perceptual similarity) stays as low as 0.10, and 98.5% of text boxes as well as all image objects remain fully editable after conversion, indicating that the system integrates reliably into standard PPT workflows used by researchers.

     **(2)**  We present a more **structured summary of the design** and core contributions of the framework in the rebuttal, making the advantages of the overall structure clearer.

3.  Reviewer **JPj1** recognizes the novelty of the task and the quality of the paper’s presentation, and raises constructive suggestions regarding data scale, annotation consistency, error-detection capability, and task generalization. In the rebuttal, we provide clarifications and additional experiments to address these points:

     **(1) For data scale**, we clarify that the number 650 refers to complete presentation documents, which together contain 13,000 real slides. The dataset used for aesthetic perception includes 6,000 labeled samples (2,000 triplets). We further discuss the data-efficiency advantages of our multi-task aesthetic model, showing that it maintains stable performance even under relatively limited annotation budgets.

      **(2) For annotation consistency**, we add detailed statistical evaluations. The ICC (Inter-Class Correlation)  values for scoring tasks exceed 0.9, and the Fleiss’ Kappa values for defect detection exceed 0.8, confirming the reliability of the annotations.

     **(3) For error detection**, we compare PresAesth with GPT-4o on 450 defective examples. PresAesth achieves an average detection rate of 96.3% and outperforms GPT-4o across all three error types. We also include a threshold stability analysis, showing that a single threshold performs robustly across multiple aesthetic tasks.

     **(4) For task generalization**, we evaluate the system on 100 documents from diverse domains and observe stable performance on aesthetic scoring, defect detection, and pairwise comparison tasks.

     **(5)** We further clarify the functional distinction between the aesthetic awareness task and the presentation generation task in the rebuttal, and refine several descriptions according to the suggestions to improve overall clarity.


We sincerely appreciate your dedication and the additional efforts you have made during this challenging period for the machine learning community. We believe that the full context of the discussion, especially the points of consensus reached, will assist in your final assessment.

Best regards,

Authors

---

### Meta-Review · Area_Chair_CDJ9 · 2026-01-09

**Summary:**

The paper introduces EvoPresent, a multi-agent framework utilizing a multi-task RL model (PresAesth) to automate academic presentation generation. The decision to accept is informed by the consensus on the work's novelty, the clear division of agent responsibilities, and the value of the constructed EvoPresent Benchmark. While reviewers initially raised concerns regarding format practicality (HTML vs. PPT), dataset scale, and dynamic evaluation, the authors provided extensive supplementary experiments (e.g., format conversion fidelity, inter-annotator agreement, and narrative coherence analysis) that effectively validated the system's robustness and utility.

**Reviewer Concerns:**

Addressed: The authors successfully provided quantitative evidence regarding the fidelity of format conversion (HTML to PPT/PDF) and clarified the inter-annotator agreement metrics (Kappa/ICC) and error detection capabilities during the rebuttal, which satisfied specific empirical questions.The rebuttal successfully addressed Reviewer SjBg's concerns regarding the practicality of HTML rendering by demonstrating high SSIM scores and editability in converted PPT/PDF files. Reviewer A3kw's concerns about the lack of dynamic evaluation were resolved through additional narrative coherence tests and a cost-benefit analysis of video iteration. Reviewer JPj1's concerns regarding data scale and task differentiation were addressed by clarifying the volume of the dataset (13,000 slides) and the distinct evaluation protocols, though the reviewer did not engage further.

Outstanding: The primary outstanding concern, highlighted particularly by Reviewer SjBg and Reviewer JPj1, is the lack of significant algorithmic novelty. The system is perceived as a "wrapper" applying standard RL and agent techniques without introducing new theoretical or architectural breakthroughs. Furthermore, despite clarifications, doubts persist regarding the sufficiency of the dataset size for robustly training models of this scale and the distinctiveness of the technical contribution beyond system engineering.

**Reviewer Scores:**

Reviewer A3kw (Score: 8) explicitly confirmed that all concerns were resolved and would maintain the strong acceptance rating. Reviewer SjBg (Score: 6) would likely maintain or slightly increase the score, as the additional data on format compatibility directly resolved their primary hesitation regarding practical utility. Reviewer JPj1 (Score: 2) would likely raise their score if they had fully participated to recognize the factual clarifications regarding dataset size and task definitions.

---

### Decision · Program_Chairs · 2026-01-26

Accept (Poster)